# Emergence of self-affine surfaces during adhesive wear

Enrico Milanese [ID] [1], Tobias Brink [ID] [1], Ramin Aghababaei[2] & Jean-François Molinari[1]

Friction and wear depend critically on surface roughness and its evolution with time. An accurate control of roughness is essential to the performance and durability of virtually all engineering applications. At geological scales, roughness along tectonic faults is intimately linked to stick-slip behaviour as experienced during earthquakes. While numerous experiments on natural, fractured, and frictional sliding surfaces have shown that roughness has self-affine fractal properties, much less is known about the mechanisms controlling the origins and the evolution of roughness. Here, by performing long-timescale molecular dynamics simulations and tracking the roughness evolution in time, we reveal that the emergence of self-affine surfaces is governed by the interplay between the ductile and brittle mechanisms of adhesive wear in three-body contact, and is independent of the initial state.

[1] Civil Engineering Institute, Materials Science and Engineering Institute, École Polytechnique Fédérale de Lausanne (EPFL), CH-1015 Lausanne, Switzerland. [2] Department of Engineering - Mechanical Engineering, Aarhus University, 8000 Aarhus C, Denmark. Correspondence and requests for materials should be addressed to J.-F.M. (email: jean-francois.molinari@epfl.ch)

The roughness of surfaces is a key parameter for all tribology-related phenomena, namely friction, wear, and lubrication. This was already clear to the pioneers of tribology, from Da Vinci[1] to Coulomb[2], who linked friction and surface morphology. Their findings were generalized in the past century by Bowden and Tabor[3], who studied the effects of adhesion and introduced the concept of real contact area. More recently, experimental evidence has shown that both natural and manufactured surfaces are self-affine over many scales[4–9]. In geophysics, the fault roughness decreases with slip[10,11] and is related to the fault's strength[12,13] and deformation mechanisms[14]. Also, for various engineering surfaces, the roughness is found to converge upon rubbing to a steady-state value[15,16]. New surfaces generated by tensile fracture are well known to be self-affine, too[8,9], and different universal Hurst exponents have been linked to different damage mechanisms[9,17]. However, the physical origins of these observations are still unclear[18].

Several theoretical surface growth models have been developed to explain roughness evolution[19,20]. Nonetheless, the generalization of simple diffusion models to the complex case of rubbing surfaces[21] still misses significant mechanisms, like the concurrence of ductile and brittle mechanisms when working the surface. Continuum numerical models are also limited, as they struggle to capture all the several intertwined non-linearities, such as contact, adhesion, plasticity, and fracture. Molecular dynamics simulations can capture the aforementioned non-linearities and atomic-scale mechanisms, but the computational cost is high[22,23] and mechanisms that take place at long time and length scales cannot be modelled.

To overcome the scale limitations in atomistic simulations, simple model potentials have recently been developed[24], which have proved to be able to capture the ductile-to-brittle transition taking place upon collision of surface asperities in adhesive wear processes[24]. When two asperities come into contact, three possible mechanisms can take place[25]: atom by atom removal in the light load and low adhesion limit[25–27], and alternatively ductile deformation[24,28,29] or brittle fracture[24,30–32] of the asperities at larger loads and moderate-to-high adhesion. Our investigations are conducted in the latter conditions, where the mechanism depends on the size of the junction formed by the two contacting asperities[24,33–35].

Here, we perform long-time molecular dynamics simulations of rubbing surfaces, investigating different initial conditions. Thanks to the adopted method, the ductile-to-brittle transition occurs spontaneously (that is, it is not enforced), permitting us to explicitly capture the debris particle formation[24,33,34] and the subsequent transition to a three-body configuration. We thus have a competition between the brittle fracture mechanism that roughens the surfaces and the ductile one that flattens them. Once the debris particle is formed, these mechanisms take place at the contact interface between the particle and the surfaces. We find that the resulting worn surfaces are self-affine and characterized by the same statistics independently of the initial state, within the investigated range. Our results also show that the debris particle wear rate is lower in the three-body configuration, i.e. after running-in.

## Results

**Simulation setup**. The investigation consists of a large set of two-dimensional (2D) molecular dynamics simulations over a long duration to maximize the chances of observing a steady state for some surface feature, as observed experimentally[15,16]. The analysed condition is dry sliding of one surface on top of the other, both described by the same model interaction potentials, at constant temperature and constant sliding velocity (see Fig. 1b

and Methods). The simulations differ from one another in bulk material properties, surface topography, temperature, and system size (see Table 1 and Methods for the full details). Throughout the article, quantities are measured in reduced units, the fundamental quantities being the equilibrium bond length $r_0$, the bond energy $\varepsilon$ at 0 K, and the particle mass $m$.

Recently, a critical length scale $d^*$ was found to govern the ductile-to-brittle transition in adhesive wear[24]: when two asperities collide, if the junction size $d$ formed by the asperities is larger than $d^*$, the asperities break and a debris particle is formed, otherwise the asperities deform plastically (Fig. 1a). The critical length scale is expressed as $d^* = \lambda \cdot \Delta w G / \tau_j^2$, where $\tau_j$ is the junction shear strength, $G$ is the shear modulus of the material, $\Delta w$ is the fracture energy, and $\lambda$ is a geometrical factor. In our simulations, the materials are described by interaction potentials, which we characterize by the maximum stress $\tau_{sf}$ on the generalized stacking fault curve at zero temperature[36,37]: the lower $\tau_{sf}$, the more ductile the material is and the larger its critical length scale $d^*$. We adopt two different initial surface morphologies: single asperity against single asperity (Fig. 1b) and self-affine surface against self-affine surface (Fig. 1e, i, m). In all cases, the initial contact takes place in a two-body configuration, that is the two surfaces come directly into contact with one another. In the single-asperity setup (Fig. 1b–d, simulations S1–S7, where S stands for single asperity, cf. Table 1), each surface is atomistically flat, except for one semicircular asperity. By sliding the top surface, its asperity collides with the asperity of the opposing surface and forms a junction. The size of the initial asperities is chosen large enough for the junction size $d$ to be greater than the critical size $d^*$ and to create a debris particle (Fig. 1c), which is then constrained to roll between the two surfaces. In the case of initially self-affine surfaces rubbing against one another (Fig. 1e–h, i–l, m–p, simulations R1 to R3, H1, G1 and G2, where R stands for rough, H for heterogeneous material, and G for grain boundaries, cf. Table 1), we construct the two surfaces with the same Hurst exponent and same root mean square roughness, but we do not control the position of the first contact, nor its size. The first contact involves several small asperities in both surfaces, which deform plastically (Fig. 1f) until they form a junction of size $d > d^*$ and the surfaces break (Fig. 1g). In this case, several asperities come into contact with the newly formed debris particle and interact with it in a brittle or ductile fashion, according to the size of the contact that is developed each time. The initial stage is even more complex when the material is heterogeneous (Fig. 1i–l, m–p, simulations H1, G1, and G2, cf. Table 1). We prepared such a case by geometrically dividing the material into irregularly shaped sub-regions (randomly distributed both in size and in position), and randomly assigning one of the two potentials within each sub-region. The two potentials differ in their critical size $d^*$, so that two different critical length scales coexist in the system. As a result of this mixture, the overall surfaces are heterogeneous in terms of inelastic behaviour (with half of the tiles being relatively more brittle than the other half). Upon asperity collision, the critical length scale for the ductile-to-brittle transition in the case of the mixture is then no longer well defined. In this case, the surfaces favour cracks within the least tough material (or at weak, heterogeneous interfaces), and plastic deformation within the toughest one (cf. Supplementary Movie 1).

After this initial stage, and independently of the original geometrical setup and of the heterogeneity of the material, the surfaces reach a state where the debris particle continuously rolls between them and works them (Fig. 1d, h, l, p). The contact now takes place in a three-body configuration, the third body being the debris particle that separates the two surfaces (i.e. the first

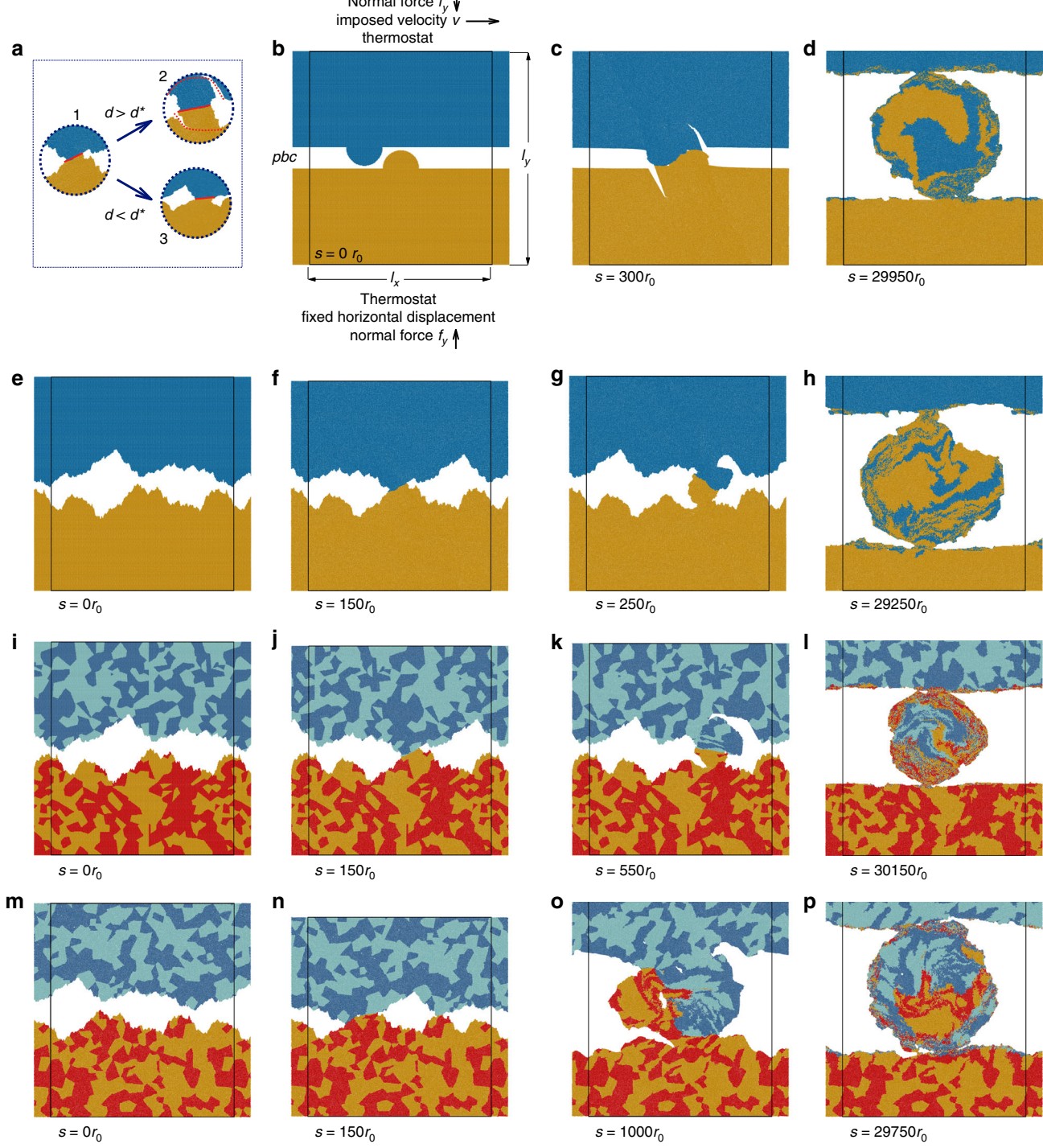

bodies). The surfaces undergo both brittle and ductile deformation and material transfer takes place both ways: from the debris particle to the surfaces and vice versa. This interplay with the third body allows for a continuous reworking of the surfaces, which appear to be self-affine whenever a steady-state roughness is reached. Remarkably, as we explain below, this self-affine morphology is independent of the initial conditions investigated. The case of heterogeneous materials is particular: each of the two materials is characterized by a different critical length scale and the effect of this on the critical length scale of the mixture is still unknown. From our results, no trace of this heterogeneity is found in the scaling of the self-affine morphology of the worn surfaces. In all cases, the inclusion of the ductile-to-brittle

transition within the modelled length scales is fundamental to capture the self-affine nature of the surfaces. When it is not included, asperity collision never leads to the formation of loose debris particles and surfaces always smoothen[38–44].

**Self-affine description.** For a complete description of the surfaces, we investigate their power spectral density (PSD) $\Phi$ per unit length, as it contains information about the contribution of all the length scales involved. Self-affine surfaces are in fact characterized by an anisotropic scale transformation[19]. This means that their heights $h(x)$ scale differently than the horizontal position $x$, and the scaling relation is[19,45] $h(\lambda x) \sim \lambda^H h(x)$, where $\lambda$ is the scaling

**Fig. 1** Ductile-to-brittle transition, simulation setup, and evolution. **a** Upon collision between two asperities (1), two possible mechanisms can take place depending on the junction size $d$: if it is larger than the critical, material-dependent value $d^*$, surfaces break and a debris particle is formed (2), else the asperities deform plastically (3). Solid red lines represent the junction of size $d$ and dotted red lines represent the crack path. **b** Setup: the two bodies are pressed together by a normal force $f_y$, while the sliding velocity is imposed on the top layer of atoms of the upper body. The bottom layer of atoms is fixed horizontally. A thermostat is applied on the layers next to the fixed boundaries. The box width $l_x$ is fixed and periodic boundary conditions are applied along $x$; the simulation cell, with initial vertical size $l_y$, is allowed to expand/shrink vertically. **b–d** Single-asperity setup, example frames from simulation S1. The point of first contact in the two-body configuration is well defined and a debris particle is formed upon asperity collision (**c**); in the three-body configuration, the debris particle wears away the surfaces while increasing in volume (**d**). **e–h** Setup with self-affine homogeneous surfaces, example frames from simulation R2. The asperities are present at all modelled scales and deform plastically upon contact in the two-body configuration (**f**) until a junction size is large enough to favour debris particle formation (**g**) and the transition to the three-body configuration (**h**). **i–l** Setup with heterogeneous self-affine surfaces: harder material is depicted in red and dark blue, softer material in yellow and light blue, example frames from simulation H1. **m–p** Setup with heterogeneous self-affine surfaces with grain boundaries, example frames from simulation G1. The steady-state surface appears rougher (**p**). In all figures, colours distinguish particles originally belonging to the top (dark and light blue) and bottom (yellow and red) surfaces; in **b–p**, black lines represent simulation box boundaries and $s$ is the sliding distance expressed in units of $r_0$

**Table 1 Summary of the simulations**

| Name | $\tau_{sf}(\varepsilon r_0^{-2})$ | $l_x$ ($r_0$) | $T$ ($\varepsilon$) | Initial $H$ (—) | Micro-structure |
|------|------|------|------|------|------|
| S1 | 3.52 | 336 | 0.075 | n/a | Single crystal |
| S2 | 3.52 | 336 | 0.050 | n/a | Single crystal |
| S3 | 3.52 | 336 | 0.025 | n/a | Single crystal |
| S4 | 3.52 | 673 | 0.075 | n/a | Single crystal |
| S5 | 3.42 | 336 | 0.075 | n/a | Single crystal |
| S6 | 3.42 | 336 | 0.050 | n/a | Single crystal |
| S7 | 3.42 | 336 | 0.025 | n/a | Single crystal |
| R1 | 3.52 | 336 | 0.075 | 0.7 | Single crystal |
| R2 | 3.52 | 336 | 0.050 | 0.7 | Single crystal |
| R3 | 3.52 | 336 | 0.075 | 0.5 | Single crystal |
| H1 | 3.52; 3.96 | 336 | 0.075 | 0.7 | Two phases |
| G1 | 3.52; 4.35 | 336 | 0.075 | 0.7 | Two phases with grain boundaries |
| G2 | 3.52; 3.90 | 336 | 0.075 | 0.7 | Two phases with grain boundaries |

S indicates simulations with initial geometry described by a single asperity on each surface. R indicates simulations with surfaces that are initially self-affine. H indicates simulations with surfaces that are initially self-affine and with heterogeneous materials without grain boundaries, modelled by even shares of the potential characterized by the two values of $\tau_{sf}$ reported. G indicates simulations with surfaces that are initially self-affine and with heterogeneous materials with grain boundaries, modelled by even shares of the potentials characterized by the values of $\tau_{sf}$ reported (see also Methods section). $l_x$ and $T$ indicate the horizontal resolution and the temperature, respectively. $H$ is the initial surface Hurst exponent for initially self-affine surfaces. Temperatures are expressed in terms of equivalent kinetic energy per atom; n/a: field not applicable to that simulation

factor, and $0 < H < 1$ is the Hurst exponent[45,46] (see Methods for a more detailed discussion about $H$). In other words, magnifying the $x$ axis by a factor $\lambda$ will produce a magnification of the heights $h(x)$ by a factor $\lambda^H$. An important consequence of this relation is that if the statistics of a self-affine surface are known at a given scale, they can be extended to all the other scales by means of the Hurst exponent $H$. The PSD of self-affine surfaces displays a power-law behaviour $\Phi(q) \sim q^{-\alpha}$ (where $q$ is the wavevector) and the Hurst exponent can be expressed in terms of the power-law exponent $\alpha$[46–48]: in the case of one-dimensional (1D) surfaces, $H = (\alpha - 1)/2$ (see Methods for more details).

The results of the analysis are displayed in Fig. 2 and Supplementary Fig. 1, which show the normalized PSD and the height–height correlation function averaged over several time steps for the worn top and bottom surfaces of different simulations, respectively. The surfaces are sampled during the steady state, where their roughness (expressed as the root mean square of heights) fluctuates around a stabilized value and their profile can be assumed to be stationary. This is the first time, to the best of our knowledge, that such a trend in the roughness evolution is numerically reproduced. It is in fact known from

experimental evidence that surfaces undergo large roughness variations in the early stage of the wear process, before settling around a steady-state value[15,49], as reproduced by our simulations (cf. Fig. 3a). The PSDs in Fig. 2a and Supplementary Fig. 1a collapse around an average value of the Hurst exponent $H = 0.7$, the lowest value being $H = 0.6$ and the highest $H = 0.8$, irrespective of the initial geometry, the material, the heterogeneity of the material, or the system size, within the range of conditions investigated. The scaling of the roughness in terms of root mean square of heights $\sigma$ with the system size is also consistent with a self-affine morphology (cf. Supplementary Fig. 2, which shows $\sigma$ for two surfaces of different size). The estimation obtained for $H$ is in good agreement with the values found for natural faults over a broad range of length scales[4,13] ($H = 0.77 \pm 0.23$), shear experiments in limestone blocks[50] ($H = 0.65 - 0.8$), and worn asphalt roads[51] ($H = 0.8$). Furthermore, the height–height correlation function $\Delta h(\delta x) = \left\langle [h(x + \delta x) - h(x)]^2 \right\rangle^{1/2}$ plotted in Fig. 2b and in Supplementary Fig. 1b allows us to observe if any crossover for $H$ is exhibited around the critical length scale $d^*$. No pronounced change of behaviour is displayed in the range of $d^*$ values investigated, and the self-affine morphology is thus the same at smaller and larger scales. The upper cutoff exhibited in Fig. 2b and in Supplementary Fig. 1b is due to the box size. The possibility of a change in the Hurst exponent at the largest investigated scales cannot be ruled out.

The fact that no change in the statistics is observed in correspondence to $d^*$ and that the surfaces are rough at all scales is possibly due to the change in the contact configuration. After the debris particle is formed, the system transitions in fact to a three-body contact configuration. The loading conditions are then changed and the critical length scale $d^*$ might thus assume a different value upon rolling contact. Furthermore, atoms can be removed at the interface by attrition, and plastic deformation also contributes to the change of the surface morphology. Another important mechanism, first put forward to explain fault roughness[10,52], is the smoothing and re-roughening of the surface by the removal of fragments from it. According to this mechanism, a fragment removed from the surface roughens it at the scale of the fragment and smooths it at larger wavelengths. In our simulations, the strong interfacial adhesion allows for this mechanism to happen ideally at all the modelled length scales. An upper length scale for the fragments is set by the current contact size between the debris particle and the surfaces. According to the described picture, larger wavelengths should be smoothed more than shorter ones, which we observe in the initial stage of the process (see Supplementary Discussion). We also remark that, beside the ductile and brittle mechanisms, surface diffusion takes place in our simulations and, without sliding, at infinite time the equilibrium surfaces would be close to atomically flat. The

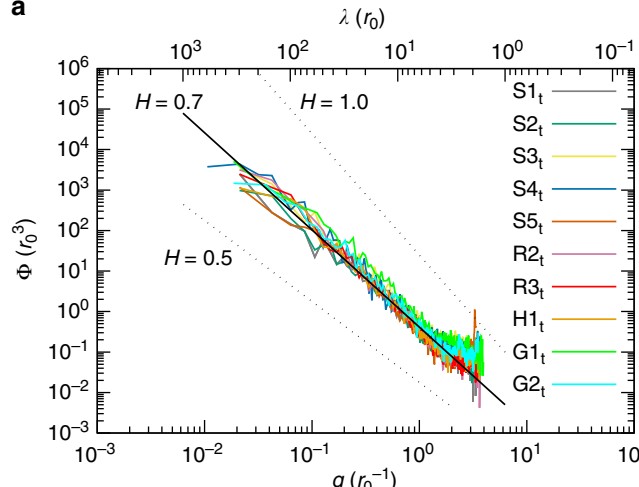

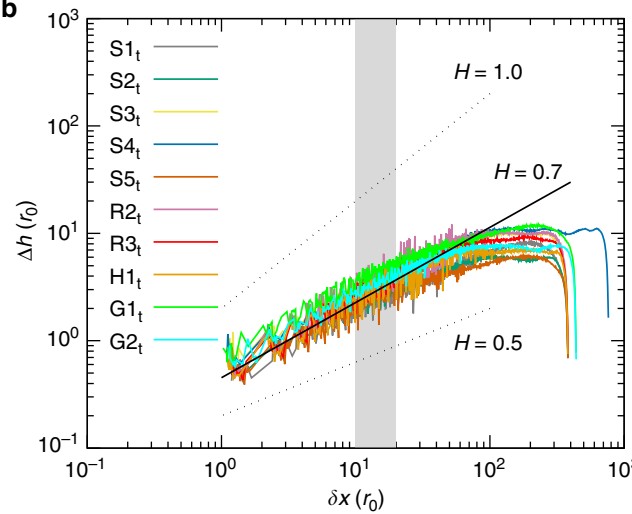

**Fig. 2** Steady-state surface morphology analysis. **a** PSD per unit length $\Phi$ as a function of the wavevector $q$ and the wavelength $\lambda$, the relation between the two being $q = 2\pi/\lambda$. **b** Height–height correlation function $\Delta h(\delta x) = \left\langle [h(x + \delta x) - h(x)]^2 \right\rangle^{1/2}$. The surfaces are taken from ten different simulations (see Table 1 for details), the subscript indicates the top surface for each simulation. Bottom surfaces for the same simulations are reported in Supplementary Fig. 1. In both **a** and **b**, the solid black straight guide-line corresponds to a Hurst exponent $H = 0.7$. Dotted black straight guide-lines show the hypothetical slope for distributions of $H = 0.5$ and $H = 1.0$. In **b**, the shaded area displays the interval of distances corresponding to the range of critical length scale values $d^*$ exhibited by the adopted potentials. No pronounced crossover is observed in the slope of $\Delta h(\delta x)$ over the range of values for $d^*$. As a consequence of the assumption of periodic surfaces, the function is roughly symmetric with respect to half the horizontal box size (hence the plateau and the following drop for large values of $\delta x$)

deformation mechanisms, though, are fast enough to counteract diffusion, and contribute to a rich distribution of the surface heights. The lattice planes are in fact not aligned across the sample surface (cf. Supplementary Fig. 3). Furthermore, when grain boundaries are modelled, each grain is initially assigned a random rotation and can rotate during the sliding process, providing an additional mechanism for the surface roughening and leading to a larger spread in the height distribution (cf. Supplementary Fig. 4).

A theoretical value of $H = 0.5$ (i.e. random correlation) for wear processes was proposed by means of a diffusion model with random deposition[21]. On the other hand, both experimental and numerical results suggest that adhesive wear is not a random Gaussian process, and that $H > 0.5$. Thus, more ingredients are needed in theoretical models that aim at describing the surface evolution during adhesive wear processes, including plastic deformation and brittle fracture. The debris particle is in contact with only a small part of the surface at every instant, localizing the deformation and the material transfer, and the sliding direction breaks down the symmetry in the evolution. This provides some similarities with the gradient percolation models used in fracture front propagation: in this class of models, the self-affine fractal front propagates towards a preferred direction (providing asymmetry) and the predicted Hurst exponent $H = 2/3$ (when small-scale effects prevail over large-scale elasticity) is consistent with our findings[20]. Models for directed polymers in a random medium also exhibit $H = 2/3$ and may provide further insights[19]. Finally, the inclusion of a scale-dependent material strength is likely another fundamental ingredient needed in theoretical models to capture a persistent Hurst exponent (i.e. $H > 0.5$)[13]. There is in fact evidence[13] that mechanical behaviour underlies a Hurst exponent $H = 0.75 \pm 0.05$ in rocks at the nanoscale.

**Evolution of a debris particle.** A particular feature of our simulations is that the two surfaces are worn mostly during three-body contact, which is relevant for the overall wear formation in both natural and industrial sliding processes[10,53]. The presence of third bodies clearly plays a key role in the emergence of the self-affine morphology and therefore we now analyse the life of a debris particle once it is formed. Details on how the debris particle is born have already been addressed elsewhere[33].

Two different geometrical setups are adopted for the simulations shown in Fig. 3: single asperity on single asperity at three different temperatures (S1–S3) and self-affine surface on self-affine surface at two different temperatures (R1 and R2). The evolution of the wear volume in Fig. 3c shows that in all cases our simulations capture both the severe wear running-in phase (formation of the debris particle, i.e. the non-zero initial wear volume) and the mild wear steady-state phase that follows. This matches experimental observations[49,54,55], as suggested by the evolution of the equivalent roughness $\sigma_{eq}$ (Fig. 3a and Supplementary Discussion).

It can be observed that the evolution of the three parameters (surface roughness $\sigma_{eq}$, tangential work $W_t$, and wear-particle volume $V$) exhibits common features among the simulations. In each simulation, a sharp increase in $\sigma_{eq}$ is observed upon formation of the debris particle, which corresponds to a sudden increase in the frictional work (inset of Fig. 3b). In this initial stage, the wear volume has been found to be proportional to the work done by the frictional force[33]: $V = W_t/\tau_j$, with $\tau_j$ being the junction shear strength and $W_t$ being the integral of the frictional force over the sliding distance. This relation applies during the particle formation, where the frictional force reaches its maximum and then slowly decreases towards small values. Over long timescales, the integral of these small values leads to a significant amount of work, which partly contributes to the particle growth. The inset of Fig. 3b and c indeed shows that the initial volume of the debris particle is larger when the frictional work is larger, and for the single-asperity setup, the frictional work appears to reach a plateau. When the original surfaces are self-affine and more collisions take place at running-in, the plateau is not clearly defined. Looking at the evolution of $W_t$ in Fig. 3b, it is clear that the frictional work is not constant over the investigated timescale, but that it keeps increasing at an

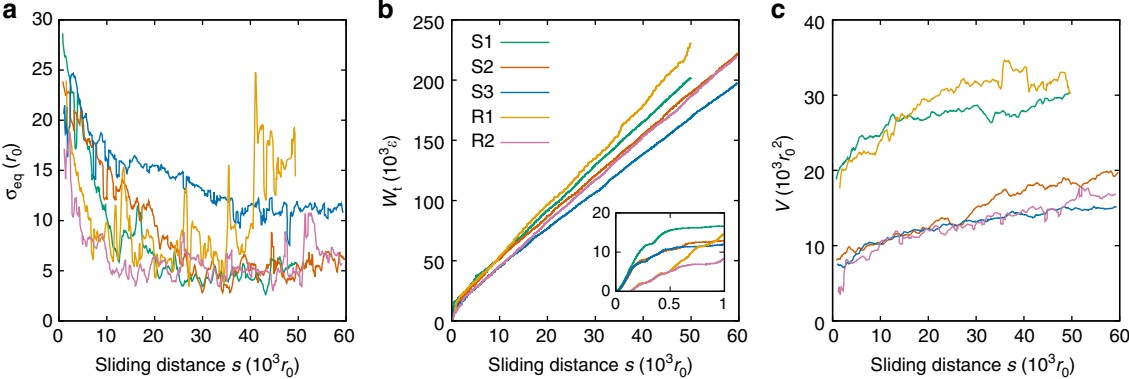

**Fig. 3** Evolution of equivalent roughness $\sigma_{eq}$, frictional work $W_t$, and wear volume $V$. See Supplementary Figs. 8 and 9 for further simulations. **a** Evolution of $\sigma_{eq}$. While for most simulations the value of $\sigma_{eq}$ stabilizes, cold temperatures (S3) and debris particle shape (R1) can inhibit this stabilization. **b** Evolution of the tangential work $W_t$ with the sliding distance. The work $W_t$ exhibits a sharp increase upon formation of the debris particle[33] (inset: $W_t$ for sliding distances up to 1000 $r_0$), after which it grows at smaller rates. **c** Evolution of the wear volume of the rolling debris particle, as defined only after its formation. In all simulated conditions, the wear rate after the debris particle formation is small compared to the ratio of the initial particle size over the sliding distance necessary to form the particle (cf. Supplementary Fig. 6), consistent with the transition from severe to mild wear[49]

approximately constant rate (consistently with a constant tangential force $F_t$, cf. Supplementary Fig. 5). The rate is nevertheless lower than the average rate displayed in the running-in stage, which governs the initial debris particle size (cf. Supplementary Fig. 6). This suggests a change in the mechanism of wear, as supported by the loss of the proportionality of the wear volume to $1/\tau_j$ after running-in (Supplementary Fig. 7) and by the change in the wear rate (Supplementary Fig. 6).

We can then split the debris particle life into two distinct phases: particle formation and particle evolution. The two situations are characterized by different wear rates (cf. Supplementary Fig. 6), the one corresponding to particle formation being much larger. This result is consistent with decreasing gouge-formation rates observed for natural faults[10]. The reduction in the friction coefficient with the sliding distance in fault lubrication processes[56] provides another consistent observation, under the reasonable assumption that wear rate and friction variations are similar under those conditions[57,58]. Likewise, shear experiments on rocks have shown vanishing wear rates in a three-body steady-state configuration[58], and wear experiments have displayed a reduction in the wear rate if the wear debris is not evacuated[53,59,60]. We ascribe the change in the wear rate, i.e. high to low, to the different contact configurations. In the initial phase, asperities belonging to both surfaces collide continuously upon sliding in a two-body configuration, and wear debris particles are repeatedly formed at a constant rate, as in Archard's picture[30]. As wear experiments are usually performed in open systems, where the wear debris particles are regularly evacuated, the formation of a full three-body contact configuration is avoided and the rate of wear particle creation is constant (due to local collisions of asperities between the two surfaces). If a three-body contact configuration develops instead (for example in faults), the third body behaves similarly to a lubricant[53], separating the first bodies. As a consequence, the two surfaces are not directly in contact with one another, but with the third body only. Wear can then take place exclusively at the interface between the third body and the surfaces. Both our simulations and the cited observations include three-body contact conditions and predict a decrease in the wear rate compatible with the introduction of a lubricating effect in the tribosystem. The rate of evacuation of debris particles from the system is thus fundamental in the evolution of surface roughness. A low evacuation rate is expected to reduce asperity collisions, which are responsible for creating roughness at scales larger than the critical junction size[24]. The mechanical behaviour

of the wear debris is also expected to affect the wear rate[53]. While in our case the debris particle has the same mechanical behaviour as the first bodies, the presence of chemical processes that harden the wear debris might actually increase the wear rate. If the debris particles favour rigid rolling over frictional sliding, though, this increase might not occur[53].

**Influence of heterogeneity**. The case of heterogeneous materials with grain boundaries (simulations G1 and G2) yields further observations. The inclusion of two different species and the presence of grains, instead of a perfect lattice, provide favourable crack propagation directions that are not present in the homogeneous cases. Cracks propagate more easily along the grain boundaries—where the mismatch of the lattices reduces the overall bond strength between the atoms at the interface—and within the bulk of the least tough species. Additionally, cracks are more likely to stop propagating when the tip meets the tougher material (cf. Supplementary Movie 1). Furthermore, the heterogeneous grains modelled in simulations G1 and G2 can be seen as inclusions of one of the two species in a matrix made of the other one, in the extreme case where the two species have the same phase fraction. The same mechanisms affecting the crack path (because of the lattice mismatch and/or different material properties) are thus expected to occur—although more localized—when the inclusions' phase fraction is smaller than the matrix phase fraction. As the latter is a case in between the homogeneous and heterogeneous bulk cases investigated here (i.e. simulations R and G, respectively), the findings of this work suggest that a similar self-affine surface morphology should be recovered. Real materials might also contain pre-existing dislocations or point defects, which are not considered in the present set of simulations. In these cases, we would expect the defects to affect the plastic response of the bulk and, thus, its critical length scale and the debris formation process. Moreover, while we restricted ourselves to one possible crystal structure, materials can also be amorphous. Assuming homogeneous materials, we would then expect the crack path to have no favourable direction because of the isotropic structure, contrary to the hexagonal lattice modelled here. Finally, surfaces undergoing continuous reworking are also known to exhibit hardening, which most likely leads to a change of the critical length scale over time, affecting in turn the wear rate.

It should be also noted that our simulations are restricted to 2D systems by the large computational demands of long-timescale simulations in three dimensions (3D). One relevant difference

between 2D and 3D systems is that in the first case, the lattices of the two bodies are more likely to locally match, allowing full, bulk-like adhesion to develop. Furthermore, in 3D, the debris particle could roll at some angle with respect to the sliding direction of the top body, as a consequence of local peaks and valleys due to the roughness along the additional dimension. It is also known that the steady-state roughness normal to the sliding direction is different than that parallel to the sliding direction[61], which cannot be captured by 2D simulations. Also, in 3D, the debris particles are not forced to pass over the same track again and again. Despite these differences, which might result in a slowdown of the roughness evolution process, we believe that the same mechanisms should be recovered in 3D systems, too, as the aforementioned evidence from experiments and field observation of faults suggests. Short timescale 3D simulations have also confirmed 2D observations for the early stages of the sliding process[33]. Further investigations are necessary in any case to address the effects of 3D geometry and of the more complex micro-structures.

## Discussion

Our molecular dynamics simulations of rubbing surfaces highlight the importance of including both ductile and brittle deformation mechanisms in the modelling of adhesive wear processes. This allows to explicitly capture the transition to the three-body configuration[24,33], where the surfaces are worn away by the debris particle.

The approach leads us to two major results. The first is the ability to track the evolution of rubbed surfaces into self-affine fractals characterized by a persistent Hurst exponent. We argue that the development of the self-affine morphology is due to smoothing and re-roughening mechanisms and that these mechanisms take place mostly in a three-body configuration, as in natural faults[10,52]. The second result is that the wear rate is lower once the system has transitioned to a three-body configuration. We ascribe this to the different contact configuration: two-body contact at running-in and three-body contact later, the latter having a lubricating effect. This unveils the role of the debris evacuation-rate in wear experiments. We conclude that accounting for the ductile-to-brittle transition in wear mechanisms is fundamental when investigating the physics of adhesive wear, from the nano- to the geological-scale.

## Methods

**Interaction potentials.** The model pair potentials used for this study belong to the family of potentials first introduced in ref. [24] and also discussed in ref. [33]. This class of model potentials allows for a critical junction size small enough to observe the ductile-to-brittle transition in adhesive wear with molecular dynamics simulations. Their main feature is to share the same elastic properties up to a bond stretch of 10% and to have a controllable yield strength by modifying the potential tail. This is made possible by modification of the Morse potential[62], leading to the expression:

$$\frac{V(r)}{\varepsilon} = \begin{cases} \left(1 - e^{-\alpha(r - r_0)}\right)^2 - 1 & r < 1.1 r_0 \\ c_1 \frac{r^3}{6} + c_2 \frac{r^2}{2} + c_3 r + c_4 & 1.1 r_0 \le r < r_{\text{cut}} \\ 0 & r_{\text{cut}} \le r \end{cases}, \quad (1)$$

where $r$ is the interparticle distance, $\varepsilon$ is the bond energy at 0 K, $r_0$ is the equilibrium bond length, and $\alpha = 3.93\ r_0^{-1}$ governs the bond stiffness. The value of $r_{\text{cut}}$ controls the cut-off distance, governing the inelastic behaviour, and the $c_i$ coefficients are chosen to ensure the potential continuity in energy and force. With respect to the values in Table 1, the potential with $\tau_{\text{sf}} = 3.96\ \varepsilon r_0^{-2}$ corresponds to the potential named P6 in ref. [24], the potential with $\tau_{\text{sf}} = 3.52\ \varepsilon r_0^{-2}$ corresponds to the potential named P4 in ref. [24], and the potential with $\tau_{\text{sf}} = 3.42\ \varepsilon r_0^{-2}$ is more ductile than potential P4 but more brittle than potential P3.

**Simulation geometry and boundary conditions.** All simulations were performed in 2D using the molecular dynamics simulator LAMMPS[63]. A simple scheme of the simulation setup is shown in Fig. 1b. Two different horizontal box sizes have been adopted, i.e. $l_x = 336\ r_0$ and $l_x = 673\ r_0$, and periodic boundary conditions are

applied in the horizontal direction. The initial vertical box size is constant ($l_y = 392\ r_0$) and the box is allowed to expand vertically, e.g. upon debris particle formation. A constant force ($f_y = 0.02\ \varepsilon r_0^{-1}$) is applied on both horizontal boundaries of the material to press the surfaces together. A constant horizontal velocity $v = 0.01 \sqrt{\varepsilon m^{-1}}$ is imposed on the first layer of atoms of the top surface. The bottom layer of atoms of the bottom surface is fixed. Temperatures are enforced by means of Langevin thermostats with a damping parameter of $0.05\ r_0/\sqrt{\varepsilon m^{-1}}$. On each body, the thermostats are applied to the three layers of atoms next to the external layer where the fixed displacement or velocity is imposed. Temperature values provided in Table 1 are expressed in terms of equivalent kinetic energy per atom. The integration is performed with a time step of $0.005\ r_0/\sqrt{\varepsilon m^{-1}}$ for a large number of steps, i.e. 1 billion for the shortest simulations and 2.6 billion for the longest one. Table 1 summarizes the main features of the simulations. Simulation names reflect the initial geometry or material heterogeneity: S stands for single asperity, R for rough self-affine surface, H for heterogeneous materials (without grain boundaries), and G for heterogeneous materials with grain boundaries. In the single-asperity setups S1–S7, we chose to model semicircular asperities, but different shapes (e.g. square or sinusoidal) are not expected to alter our findings[24]. In simulations H1, G1, and G2, both bodies are modelled by two phases (using two different potentials), present in equal shares and randomly distributed in 500 tiles generated from 500 points randomly distributed in the simulation cell by means of Voronoi tessellation. No grain boundaries between the tiles are modelled in simulation H1. Grain boundaries are modelled in simulations G1 and G2: each of the 500 tiles is a grain, whose orientation is determined by a random rotation, leading to lattice mismatch. In simulations G1 and G2, one of the two potentials is the same used for simulations S1, S2, S3, S4, R1, R2, and H1, and it is characterized by $\tau_{\text{sf}} = 3.52\ \varepsilon r_0^{-2}$ (cf. Table 1). The other adopted potential has the same equilibrium energy $\varepsilon$, but its equilibrium and cut-off distances are scaled by a factor 0.90 in simulation G1 (resulting in $\tau_{\text{sf}} = 4.35\ \varepsilon r_0^{-2}$) and 0.95 in simulation G2 ($\tau_{\text{sf}} = 3.90\ \varepsilon r_0^{-2}$). The potential well of the cross-terms between the two species is $0.9\varepsilon$; its equilibrium and cut-off lengths are given by the average of the respective lengths of the two species. The starting system is obtained by constructing a bulk micro-structure, heating it up and then annealing it at the target temperature, allowing the grains to reach an equilibrium configuration. Atoms are then removed from the system based on a purely geometric criterion to obtain two distinct rough surfaces.

**Self-affine surfaces.** Fractal surfaces whose heights $h(x)$ scale differently than the horizontal distance $x$ are self-affine fractals, and they obey the scaling relation[19,45] $h(\lambda x) \sim \lambda^H h(x)$, where $\lambda$ is the scaling factor and $H$ is the Hurst exponent, which is $0 < H < 1$ for fractional Brownian motion (fBm)[46]. The Hurst exponent describes the correlation between two increments in the surface. Assuming $x_1 < x_2 < x_3 < x_4$, let us consider the two height increments $\Delta h_1 = h(x_2) - h(x_1)$ and $\Delta h_2 = h(x_4) - h(x_3)$. For $H = 0.5$, $\Delta h_1$ and $\Delta h_2$ are randomly correlated (i.e. standard Brownian motion), which means that $\Delta h_2$ has a 50% probability of having the same sign of $\Delta h_1$. For $0 < H < 0.5$, the increments $\Delta h_1$ and $\Delta h_2$ are negatively correlated, that is $\Delta h_2$ is more likely to have the opposite sign of $\Delta h_1$ (the motion is anti-persistent: a positive increment is more likely to be followed by a negative one). Finally, for $0.5 < H < 1$, the increments are positively correlated, that is $\Delta h_2$ is more likely to have the same sign of $\Delta h_1$ (the motion is persistent: a positive increment is more likely to be followed by another positive one). The generation of engineering surfaces, that is surfaces that are manufactured, is known to be non-stationary and random[5], and it can be described as a non-stationary process with stationary increments, which allows to relate the fractal dimension $D$ of the surface with its Hurst exponent $H$ through its Euclidean dimension $n$[46]: $D + H = n + 1$. Moreover, the statistics of this class of surfaces have been investigated studying the Weierstrass–Mandelbrot function[47], allowing to relate the fractal dimension $D$ and the power law exponent $\alpha$ of the PSD of a 1D surface profile[47,48] as $\alpha = 5 - 2D$. Under these assumptions, $H = (\alpha - 1)/2$ for a 1D surface ($n = 1$).

The latter allows to estimate the Hurst exponent $H$ from a linear fit of the data in the log–log plane, discarding the tail ($\lambda < 4\ r_0$), where the definition of surface for a discrete system breaks down and data are inevitably polluted by the numerical surface reconstruction. This method has been shown[64] to be accurate but also to be affected by a systematic error that can possibly lead to an underestimation of $H$. This underestimation decreases by increasing the system size, and for system sizes of the order of magnitude investigated in this paper it is at most 0.1. The range of values of $H$ in our study would then be $H = 0.7$–$0.9$. We refer throughout the text to the values found with the fit, without correcting for the underestimation, for consistency with the methods adopted in the literature (e.g. refs. [4,6,13]), where no correction is applied.

**Spectral analysis.** Let us consider a surface of length $L$, whose height is defined by the continuous function $h(x)$, where $x$ is a spatial coordinate. We refer to the PSD of such a surface in terms of PSD per unit length $\Phi_h(q)$, $q$ being the wavevector, defined as[65]

$$\Phi_h(q) \equiv \frac{1}{L} \left| \int_L h(x) e^{-iqx} dx \right|^2, \quad (2)$$

where the integral is the continuous Fourier transform of $h(x)$. The PSD defined in Eq. (2) is equivalent to the PSD of a continuous function $h(x)$ that is zero

everywhere except over a distance $L$, normalized by $L$. (Note that the specification 'per unit length' is often dropped in surface roughness analyses[55,66].) In particular, we estimate $\Phi_h$ as

$$\Phi_h(q_n) \approx \Delta x \cdot P_h(q_n), \tag{3}$$

where $P_h(q_n)$ is the classical periodogram[65,67]:

$$P_h(q_n) = \frac{1}{N} \left| \sum_{k=0}^{N-1} h_k e^{-iq_n x_k} \right|^2, \tag{4}$$

the summation being the discrete Fourier transform of the surface. In fact, $h(x)$ is known only at a discrete set of $N$ points $x_k$ ($k = 0, 1, \ldots, N - 1$), regularly sampled at an interval $\Delta x$, such that $h_k = h(k\Delta x)$ are the known values of $h(x)$. In our case, $\Delta x = L/N$, $N$ being the number of atoms belonging to a surface of length $L$, and is approximately 1 $r_0$.

It can be shown that both the PSD $\Phi_h$ per unit length and the periodogram $P_h$ are normalized such that $\int_q \Phi_h(q) dq$ and $\sum_n P_h(q_n)$ are equal to the mean squared amplitude $\sigma^2$ of $h(x)$ and $h_k$, respectively.

Note that the derivative of both $P_h$ and $\Phi_h$ is the same in the log–log plane, as their estimation differs only by a multiplicative factor $\Delta x$ (see Eq. (3)): the estimated self-affine exponent does not change if one or the other is considered.

**Height–height correlation analysis**. Another suitable method to estimate the Hurst exponent $H$ is to investigate the height–height correlation function[19], which describes the change of heights $\Delta h$ between two points at distance $\delta x$ horizontally:

$$\Delta h(\delta x) = \left\langle [h(x + \delta x) - h(x)]^2 \right\rangle^{1/2}, \tag{5}$$

where the angle brackets indicate spatial average. The height–height correlation function scales as $\Delta h(\delta x) \sim \delta x^H$: it is therefore possible to determine the Hurst exponent $H$ from its log–log plot. It is also possible to observe potential upper and lower cut-off values of $\delta x$ limiting the scaling regime.

**Average roughness quantification**. The surface roughness is defined as the variations in height of the surface profile with respect to an arbitrary plane of reference[55], which in our case is always taken to be the mid-plane of the surface heights. To quantify the surface roughness, several parameters are used—in particular in engineering practice—but we limit our discussion to the root mean square of heights $\sigma$, which in the case of a surface discretized in a set of $N$ points is given by

$$\sigma = \sqrt{\frac{1}{N} \sum_{k=1}^{N} h_k^2}, \tag{6}$$

where $h_k$ is the distance of the point $k$ from the plane of reference. When the system as a whole is investigated, the equivalent $\sigma$ of the composite surface (given by the two surfaces of the top and bottom materials) becomes[55]

$$\sigma_{\mathrm{eq}} = \sqrt{\sigma_{\mathrm{top}}^2 + \sigma_{\mathrm{bottom}}^2}. \tag{7}$$

The root mean square of heights $\sigma$ can also be expressed as a function of the zeroth moment of the PSD per unit length $\Phi_h$, assuming that the surface profile is continuous[6,65]:

$$\sigma^2 = \int_{q_l}^{q_h} \Phi_h(q) dq, \tag{8}$$

where $q_l$ and $q_h$ are the lowest and highest wavevectors modelled in the system. If $q_h \gg q_l$, it has been shown[6] that $\sigma^2 \propto q_l^{-(4-2D)}$.

**Data analysis**. All frames are visualized with OVITO[68]. Due to the long duration of the simulations, frames are stored every 1,000,000 steps. At first, a surface is reconstructed by identifying atoms with low coordination number. This preliminary surface includes (i) the top and bottom surface portions that are not in contact with the debris particle and (ii) the debris particle surface portion that is not in contact with any surface. The two top and bottom surfaces are then reconstructed, their portion in contact with the debris particles being approximated by the closest group of atoms belonging to a straight segment. This approximation to a straight segment gives no loss of information for the spectral analysis by means of the Fast Fourier Transform up to high wavevectors[69]. The atoms contained between the two segments and the previously identified debris particle surface identify the debris particle and are thus discarded in any top and bottom surface analysis. Each reconstructed surface now consists of irregularly spaced atoms and is processed independently. The atom positions are linearly interpolated to recreate a bijective profile that is then discretized in $N$ points, evenly spaced along the horizontal $x$-axis, where $N$ is the number of unevenly spaced atoms belonging to the surface prior to the linear interpolation. This step allows to proceed with an analysis of the surface by means of the classical periodogram through a Fast Fourier Transform algorithm, as the Lomb–Scargle periodogram for unevenly spaced data is known to provide poorer results in the spectral analysis of surface morphologies[69,70]. The horizontal spacing of the re-sampled surface is

approximately 1 $r_0$. Forcing a bijective profile results in a noisy geometry where overhangs are present, affecting the large wavevector amplitudes, but does not alter the data at lower wavevectors, where the self-affine morphology is observed. The interval between two consecutive time steps used for averaging the PSD is large enough for the particle to have rolled over the whole surface at least one time.

The data for $\sigma$ and $\sigma_{\mathrm{eq}}$ are averaged over 10 consecutive data points. The tangential work $W_t$ is computed as the integral of the tangential force $F_t$ over the sliding distance $s$: $\int F_t ds$; the tangential force values are stored every 5000 steps and are averaged over 2000 data points. The debris particle volume $V$ is computed by multiplying the number of atoms belonging to the debris particle and the atomic volume. The latter is computed as the lattice unit cell volume at the temperature $T$, divided by the number of atoms in the unit cell. As the detection code is designed for the most common situation of the particle being in contact with both surfaces at the same time, due to the particular geometry at some time steps, e.g., when the particle is only in contact with one of the two surfaces and is not rolling, the data for the surface roughness $\sigma$ and $\sigma_{\mathrm{eq}}$, the wear volume $V$, and the linear fit exponent $\alpha$ may be unevenly spaced locally around some time steps.

## Data availability
All raw data supporting the findings of this study are available from the authors upon request.

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

## Acknowledgements

The authors gratefully acknowledge D. Bonamy and A. Dubois for the helpful discussion on surface growth models. This work was supported by EPFL through the use of the facilities of its Scientific IT and Application Support Center.

## Author contributions

E.M. performed the numerical simulations and the analyses of the data. E.M., T.B., R.A., and J.-F.M. participated in the design of the simulations, the discussions, and the manuscript preparation.
