## [Peer Review File · Nature Communications]

Reviewers' comments:

Reviewer #1 (Remarks to the Author):

This manuscript explores the spontaneous formation of self-affine surfaces through wear in friction, by means of MD simulations.

The authors take a bold step in exploring this multiscale challenge, facing a numerically intensive approach. Their choice to restrict to 2D is appropriate and well motivated, although it leaves questions open with regards to the likely anisotropic statistical properties of the resulting worn surfaces in real-life 3D geometry.

Overall, this paper addresses an interesting issue with appropriate state-of-the-art methods and statistical data analysis. The results are reported using a precise language, constructing an overall well-organized presentation.

I have a few criticisms which should be addressed before publication:

* The choice of setting the Hurst exponent of the initial configuration of both R1 and R2 simulations to 0.7 (caption of Table E1) is quite convenient for speeding up the approach to the steady state. It is less so to convince the reader of one of the main claims of this paper, namely that the system approaches the same steady state with universal properties not depending on the initial state. I think the paper would gain substantial credibility if a R3 simulation was carried out starting from some initial state with significantly different statistical properties, e.g. $H=0.5$, leading eventually to the same type of surface statistics.

* This work is clearly about out-of-equilibrium dynamics, for which the authors address the approach to a dynamical steady state, and the statistical properties of the steady state itself. In several passages in the paper the authors refer to it correctly as "steady state", but occasionally (e.g. lines 15, 188, 225) they refer to it as "equilibrium". This might lead to confusion and should be amended, because "equilibrium" is a technical term reserved for a situation where no time-dependent external forces or fields act on a system that has been left alone to interact weakly with some huge "environment" for a very long time.

* The self-affine description in the text (lines 158-175) is clear enough. I feel that the repetition, especially lines 444-450, is useless and could be safely deleted.

* In the standard physics literature (as in, e.g., Aschroft-Mermin's book), the symbol $[\text{Greek}] \omega$ and the term "frequency" are both normally reserved to the Fourier-conjugated variable to time, the one that is measured in Hz or rad/s in the SI.

The Fourier-conjugated variable to position x is rather a "wave vector", with standard symbols " k " and/or " q ", measured in m^{-1} in the SI, and in r_0^{-1} in model units, as in the horizontal axis of Fig. 2. As the authors chose the letter k for the indexes of a vector of surface positions, I recommend adopting q for this quantity, and amend the paper throughout.

* In particular the unnumbered equation for PSD after line 464 should have the independent variable "omega" (renamed "q") also at the right side, or it makes little sense. More specifically, FFT is just a popular algorithm to compute the Fourier transform of a data series, but the results would not change if a different algorithm was used. This equation would become far clearer if the explicit formula of the FT with the precise adopted normalization was indicated at the right side.

* Related to the previous point, in the reconstruction of the two surfaces described in lines 497-501 the authors should usefully report what lateral resolution they adopt for the x_k discretization and how they proceed in the event of surface overhangs.

* The verb "dump" used at lines 495 and 516 literally means "throw away", e.g. something in a garbage bin, and so is understood by anybody familiar with English language and unfamiliar with lammps jargon. I recommend some more standard verb such as "save" or "record".

In summary, the manuscript addresses an important question and reaches the quality standards for publication in Nature Communications. It may significantly improve with a minor revision addressing the points raised above.

Reviewer #2 (Remarks to the Author):

Milanese et al

The paper by Milanese et al presents extensive simulations of the process of wear between surfaces with roughness and shows how the surface topography evolves. The simulations are impressive and novel due to the long time scale over which they are conducted, allowing the surfaces to reach (in most cases) a steady-state which can then be compared to experimental results. The paper provides, for the first time, an explanation for the very surprising observation that surfaces subjected to dry sliding often evolve to form self-affine roughness with a Hurst exponent in the range of 0.7. This has been seen both for natural surfaces like faults and for engineering surfaces subjected to sliding conditions in lab tests or in functioning mechanical devices. The authors see that the behavior is robust and occurs independent of the initial surface roughness (be it a single bump or a self-affine surface to begin with), or with including heterogeneous mechanical properties by considering a two-phase material. The problem of how surfaces evolve is extremely important and broad, with implications for understanding and predicting earthquakes, and for designing and predicting the friction, wear, and lifetime of engineered systems like vehicles, machinery, and manufacturing processes.

I find that this work represents a potential breakthrough since an explanation for the persistence of self-affinity in worn surfaces with a rather specific range of Hurst exponents has been a mystery, and the processes uncovered to explain it can lead to predictive understanding of real practical importance. The work is carefully performed and clearly presented.

I have a few concerns for the authors to address before being convinced this is suitable for publication in Nature:

1. On page 1 the authors state, "The inclusion of the ductile-to-brittle transition permits us to explicitly capture the debris particle formation." I believe that the ductile-to-brittle transition

emerges naturally from the simulation and is not imposed at a certain contact size, but the sentence above suggests the latter. Please clarify how the ductile-to-brittle transition is managed; does it occur 'on its own' or is it enforced into the simulation?

2. On page 2, lines 167-169, the authors discuss the important distinction between surfaces with Hurst exponents <0.5 and >0.5 . The description is unclear as presented and could be confusing to the broader audience for this venue. In particular, the authors should clearly explain what is meant by "increments." Furthermore, what is meant by "increments are positively/negatively/randomly correlated"?

3. The authors do not include grain boundaries, pre-existing dislocations, or defects like inclusions into their simulations. Nor do they consider amorphous materials, which are often found when oxides are formed and when materials have been subjected to large and rapid strains, such as in faults. The one complex case they consider is a 50/50 mixture of two phases, which does show behavior consistent with the behavior of the perfect single crystals they simulate. However, perfect single crystals are highly exceptional cases. I would be more strongly convinced that this work should be published in Nature if they simulated cases with a realistic population of such defects. At the very least, a discussion of these points with reference to other studies or to physically-based arguments to provide more understanding would help.

4. Related to the above point, materials subjected to long periods of loading can exhibit strain hardening and fatigue failure. These effects can have significant consequences for wear. The authors should at least discuss this point and the effect these phenomena would have if they were included in the simulations.

5. The authors do discuss the fact that they made the choice of restricting the simulations to 2D systems in order to carry the simulations out to long time scales. This choice is understandable, but further discussion of the limitations of this are needed. The authors discuss possible consequences on page 6, lines 347-357. They point out that that "In real 3D tribosystems full adhesion might not take place." The referee does not understand the point being made here; please clarify. The authors then state, "and the debris particles are not forced to pass over the same track again and again, possibly affecting the process as described above" (referring to the evacuation of debris). Even in 3D, there would not be any evacuation if two semi-infinite plane surfaces were simulated. However the referee agrees that particles would find locations where little to no contact occurs in 3D, which is not possible in 2D. The authors should discuss the limitations imposed by the 2D constraint in somewhat more depth. Their only statement is "Despite these differences, which might result in a slow down of the roughness evolution process, we argue that the rationale put forward should be recovered in 3D systems too, as the aforementioned evidence from experiments and field observation of faults suggests". The referee agrees a slow down of wear may occur for 3D vs. 2D simulations, but what other differences could be expected, including those resulting from the different nature of contact stresses for 3D vs. 2D systems? Strengthening this discussion further would be helpful. Could the authors run a short 3D simulation to at least compare what happens at the early time scales to see if there is some degree of commonality between 2D and 3D? That would strengthen the arguments of applicability for this paper.

6. It would further strengthen the paper if the authors could consider a finite contact geometry, e.g. a ball on a flat, where debris evacuation can occur. If similar trends are seen, then it would further strengthen the general applicability of the paper.

7. A minor point: in the conclusion, the authors state that their observation of a change in the wear rate contradicts Archard's law and argue that this could be resolved if evacuation of debris were included. However, in macroscopic tests, changes in wear rate are often seen – high wear occurs during the 'run in' phase in the first few tens-100's of cycles of most lab tests for a wide range of contact conditions. A steady wear rate following Archard's law is often applied after this first transient phase. The fact that the authors see a change in behavior in the early stages of their simulation is consistent with many observations of wear.

In summary, while this work considers a much longer time scale than any other wear simulations with this level of resolution, and even though a number of different conditions are varied, the other limitations of the work constrain the general applicability of the very interesting results. The

referee is optimistic that the mechanism of self-affine surface topography generation the authors uncover would persist and be found in further simulations and thus a set of results that show this would firmly justify the paper's acceptance for publication in Nature. As it stands, the paper is still a very strong and innovative contribution, and at the very least is worth publication in Nature Materials.

Reviewer #3 (Remarks to the Author):

Milanese and co-workers study the evolution of surface roughness during wear in a specific coarse-grained particle model developed by a subset of the authors in previous publications (Nature Comm. 7, 11816 (2016); PNAS 114, 7935 (2017); Phys. Rev. Lett. 120, 186105 (2018)). The calculations are identical to these previous works but the interpretation of the data is new. The authors claim that the surfaces that develop in their model are self-affine with a Hurst exponent of roughly 0.7, in agreement with many experimental findings, but I have doubts regarding this interpretation. While the overall subject matter is interesting, based on the model and interpretation of the outcome I do not recommend publication of this manuscript in Nature Communications. More detailed criticism follows.

Model

The "coarse-grained" model used by the authors cannot correspond to any physical system. It looks like an ad-hoc interatomic potential (that would then not be coarse-grained), but it is certainly not a systematic coarse-graining of some underlying atomic-scale model as is carried out in for example the polymer community. There are several inconsistencies that are (because of this) associated with the coarse-graining idea: Their model shows dislocation plasticity, but the scale of the particles is certainly larger than an atom (otherwise it wouldn't be coarse-grained). Dislocation as the carrier of plastic deformation here are not the mechanism to be expected at these coarse-grained scales. Similarly, the authors carry out their simulations at finite temperature but fluctuations decrease with coarse graining-level and may have no significance here. No systematic attempt to clarify lattice defects in the coarse-grained description and scaling of fluctuation with coarse-graining level. In addition, the model is two-dimensional and this may affect what the authors are seeing (e.g. the elastic response has a logarithmic, not a $1/r$, singularity; melting and therefore the effect of temperature is different in 2D and 3D), although my feeling is that this may not affect the overall results seen by the authors.

Numerical studies of model systems are useful, but those models should be clearly guided by physical reality and the farther the model is away from physical reality the less interesting I would judge the obtained outcome for a broad readership such as that of Nature Communications. I acknowledge that the fact that results obtained with this model have appeared in Nature Comm., PNAS and Phys. Rev. Lett. and that not all of my colleagues may share this view on the model.

Roughness

The authors quantify surface roughness in their calculations. They find a power-spectral density and height-difference autocorrelation function indicative of self-affine scaling. Scaling exponent is not clear. The authors claim $H=0.7$ but $H=0.5$ would equally well be possible given the spread of their data. (E.g. S1b is very close to 0.5.) The only clear conclusion to be drawn from PSD and height correlation is $0.5 < H < 1.0$ which is in the range of what is found experimentally.

The images shown Fig. 1h and 1l illustrate that their surfaces are very flat but exhibit a series of steps that, judging from Fig. 3, lead to an rms amplitude of about five steps and a plateau width of about 20 particles. In other words, the height distribution is hardly continuous or a Gaussian. I find it hard to call these discrete surfaces "self-affine". If I scale them down (according to the

procedure described by the authors) this reduces the plateau width and clearly leads to surfaces that look different; the notion of self-affinity for these discrete systems studied by the authors would need either rougher surfaces or larger areas (i.e. a continuum-like height distribution).

It is also worth noting that the Fourier transform of a step gives something that looks like Hurst exponent 0.5. The height correlation of a step shows something similar. I invite the author to generate a line profile with 5-10 random steps and compute PSD and height correlation. It looks almost exactly like what is presented in this paper. I therefore believe the "self-affine scaling" reported by the authors is an artifact of their analysis.

Junction size

The authors refer to an energy-balance criterion similar to Griffith's for fracture that I believe goes back to Rabinowicz when talking about wear particle creation. They express this criterion in terms of what they call the critical junction size. This "junction-size" d was introduced in an earlier paper (Nature Comm. 7, 11816 (2016)) but appears to be an ill-defined and rather arbitrary concept. If I look at Fig. E6 then it is not clear to me why the junction size is determined by the red lines indicated by the authors and not by the shortest paths across the particle which is clearly smaller. There appears to be no clear definition of this junction size.

According to the authors, the deformation should be plastic below a critical d^* but this is inconsistent with their own simulations. Once a particle has formed, the junction size (the point of contact between particle and surface) is very small and the deformation should be plastic, but no such plastic deformation appears to be observed. However, the authors don't really need to refer to the junction size model for interpretation - I'm not sure why it is in this paper at all.

Other points

* I don't think Da Vinci, Amontons and Coulomb were aware of the importance of surface roughness. This is typically attributed to Bowden & Tabor, but the earliest reports date to the end of the 19th century beginning of the 20th century.

* What is meant by "different universal dimensionalities" in the introduction? I suppose the authors mean Hurst exponents or fractal dimensions?

* Model is called coarse-grained and initially then later referred to as atomistic.

* What does the Boltzmann constant k_B mean in the context of the coarse-grained model and what is its value, i.e. how is temperature measured? Did the authors define some reference temperature, e.g. the melting temperature of the solid? (Melting in 2D is very different than in 3D so this may not be very meaningful.) See also my comment above concerning temperature and coarse-graining.

* If the authors use a standard Langevin thermostat, then their friction force will be the damping force of the Langevin because the Langevin thermostat contains reference velocity which is zero by default. How did the author do the thermalization exactly?

* The authors say fractional Brownian motion has non-stationary increments. This is wrong. The increment process (fractional Gaussian noise) is stationary.

* What does the sentence "even if two different critical length scales coexist" mean?

* How did the authors extract the roughness? I guess it was mapped onto a grid (of which spacing?) before carrying out the Fourier transform. How were the wear particles that are still

attached to a surface at the end of the calculation removed?

* FFT(z) should be specified explicitly because normalization factors can vary. Indeed FFT is an algorithm. I believe the authors mean the discrete Fourier transform.

* Fig. 3 should have a scale bar.

* As already stated above, it would be interesting to see height distributions.

We are grateful to all three reviewers for their comments and questions. We have taken all of them into account and in the process we believe that the paper’s clarity has greatly improved. In the following, the reviewers’ remarks are reported in italic, followed by our response in standard text. Please note that following the manuscript revision, the Figure numbering has changed, and in our response we refer to the new numbering. Figures and tables in this response are numbered as R.X (e.g. Figure R.1).

1 Reviewer 1

This manuscript explores the spontaneous formation of self-affine surfaces through wear in friction, by means of MD simulations.

The authors take a bold step in exploring this multiscale challenge, facing a numerically intensive approach. Their choice to restrict to 2D is appropriate and well motivated, although it leaves questions open with regards to the likely anisotropic statistical properties of the resulting worn surfaces in real-life 3D geometry.

Overall, this paper addresses an interesting issue with appropriate state-of-the-art methods and statistical data analysis. The results are reported using a precise language, constructing an overall well-organized presentation.

I have a few criticisms which should be addressed before publication:

** The choice of setting the Hurst exponent of the initial configuration of both R1 and R2 simulations to 0.7 (caption of Table E1) is quite convenient for speeding up the approach to the steady state. It is less so to convince the reader of one of the main claims of this paper, namely that the system approaches the same steady state with universal properties not depending on the initial state. I think the paper would gain substantial credibility if a R3 simulation was carried out starting from some initial state with significantly different statistical properties, e.g. $H=0.5$, leading eventually to the same type of surface statistics.*

We thank the reviewer for the suggestion. We have performed an additional simulation R3 with an initial Hurst exponent $H = 0.5$. Remarkably, the steady-state top and bottom surfaces appear to be self-affine with an Hurst exponent H around 0.7 as observed for the already investigated surfaces. The manuscript has been updated to include the new results.

** This work is clearly about out-of-equilibrium dynamics, for which the authors address the approach to a dynamical steady state, and the statistical properties of the steady state itself. In several passages in the paper the authors refer to it correctly as "steady state", but occasionally (e.g. lines 15, 188, 225) they refer to it as "equilibrium". This might lead to confusion and should be amended, because "equilibrium" is a technical term reserved for a situation where no time-dependent external forces or fields act on a system that has been left alone to interact weakly with some huge "environment" for a very long time.*

We agree and thank the reviewer for catching this: the use of the term ‘equilibrium’ is indeed an oversight. The manuscript has been updated accordingly.

** The self-affine description in the text (lines 158-175) is clear enough. I feel that the repetition, especially lines 444-450, is useless and could be safely deleted.*

To avoid repetitions, we have removed the discussion on H from the main part, and moved it to the Methods section, part of which has been extended to address a comment from Reviewer 2.

** In the standard physics literature (as in, e.g., Aschroft-Mermin’s book), the symbol $[\text{Greek}]\omega$ and the term "frequency" are both normally reserved to the Fourier-conjugated variable to time,*

the one that is measured in Hz or rad/s in the SI. The Fourier-conjugated variable to position x is rather a 'wave vector', with standard symbols 'k' and/or 'q', measured in m^{-1} in the SI, and in r_0^{-1} in model units, as in the horizontal axis of Fig. 2. As the authors chose the letter k for the indexes of a vector of surface positions, I recommend adopting q for this quantity, and amend the paper throughout.

We understand and agree. Figure 2a now contains two x axes, showing both wavevectors and wavelengths. The terminology and notation have been updated throughout the manuscript.

** In particular the unnumbered equation for PSD after line 464 should have the independent variable "omega" (renamed "q") also at the right side, or it makes little sense. More specifically, FFT is just a popular algorithm to compute the Fourier transform of a data series, but the results would not change if a different algorithm was used. This equation would become far clearer if the explicit formula of the FT with the precise adopted normalization was indicated at the right side.*

We agree that we over-simplified this part of the Methods section. We have therefore expanded it, explaining how the discrete estimation relates to the continuous one and showing the adopted normalization. Because of this, we changed the notation for the adopted PSD estimation and therefore updated the text and figures labels accordingly.

** Related to the previous point, in the reconstruction of the two surfaces described in lines 497-501 the authors should usefully report what lateral resolution they adopt for the x_k discretization and how they proceed in the event of surface overhangs.*

We expanded the *Data analysis* paragraph of the Methods section accordingly, including also remarks from Reviewer 3.

** The verb "dump" used at lines 495 and 516 literally means "throw away", e.g. something in a garbage bin, and so is understood by anybody familiar with English language and unfamiliar with lammgs jargon. I recommend some more standard verb such as "save" or "record".*

We agree and updated the manuscript accordingly.

In summary, the manuscript addresses an important question and reaches the quality standards for publication in Nature Communications. It may significantly improve with a minor revision addressing the points raised above.

2 Reviewer 2

Milanese et al

The paper by Milanese et al presents extensive simulations of the process of wear between surfaces with roughness and shows how the surface topography evolves. The simulations are impressive and novel due to the long time scale over which they are conducted, allowing the surfaces to reach (in most cases) a steady-state which can then be compared to experimental results. The paper provides, for the first time, an explanation for the very surprising observation that surfaces subjected to dry sliding often evolve to form self-affine roughness with a Hurst exponent in the range of 0.7. This has been seen both for natural surfaces like faults and for engineering surfaces subjected to sliding conditions in lab tests or in functioning mechanical devices. The authors see that the behavior is robust and occurs independent of the initial surface roughness (be it a single bump or a self-affine surface to begin with), or with including heterogeneous mechanical

properties by considering a two-phase material. The problem of how surfaces evolve is extremely important and broad, with implications for understanding and predicting earthquakes, and for designing and predicting the friction, wear, and lifetime of engineered systems like vehicles, machinery, and manufacturing processes.

I find that this work represents a potential breakthrough since an explanation for the persistence of self-affinity in worn surfaces with a rather specific range of Hurst exponents has been a mystery, and the processes uncovered to explain it can lead to predictive understanding of real practical importance. The work is carefully performed and clearly presented.

I have a few concerns for the authors to address before being convinced this is suitable for publication in *Nature*:

1. On page 1 the authors state, “The inclusion of the ductile-to-brittle transition permits us to explicitly capture the debris particle formation.” I believe that the ductile-to-brittle transition emerges naturally from the simulation and is not imposed at a certain contact size, but the sentence above suggests the latter. Please clarify how the ductile-to-brittle transition is managed; does it occur ‘on its own’ or is it enforced into the simulation?

Thank you for your encouraging comments. Indeed, we do mean that the ductile-to-brittle transition occurs naturally, i.e. on its own, and we do not enforce it into the simulation. We have reformulated the sentence in the manuscript to clarify this point.

2. On page 2, lines 167-169, the authors discuss the important distinction between surfaces with Hurst exponents <0.5 and >0.5 . The description is unclear as presented and could be confusing to the broader audience for this venue. In particular, the authors should clearly explain what is meant by “increments.” Furthermore, what is meant by “increments are positively/negatively/randomly correlated”?

We now refer the reader to the Methods section for a more detailed explanation, as Reviewer 1 suggested that the explanation in the Results section was sufficient (and the one in the Methods section redundant).

3. The authors do not include grain boundaries, pre-existing dislocations, or defects like inclusions into their simulations. Nor do they consider amorphous materials, which are often found when oxides are formed and when materials have been subjected to large and rapid strains, such as in faults. The one complex case they consider is a 50/50 mixture of two phases, which does show behavior consistent with the behavior of the perfect single crystals they simulate. However, perfect single crystals are highly exceptional cases. I would be more strongly convinced that this work should be published in *Nature* if they simulated cases with a realistic population of such defects. At the very least, a discussion of these points with reference to other studies or to physically-based arguments to provide more understanding would help.

4. Related to the above point, materials subjected to long periods of loading can exhibit strain hardening and fatigue failure. These effects can have significant consequences for wear. The authors should at least discuss this point and the effect these phenomena would have if they were included in the simulations.

We thank the reviewer for these comments. We have performed two additional simulations of mixtures with grain boundaries (named G1 and G2). In both cases two different species are modelled in the system. One of the two potentials is the same used for simulations S1, S2, S3, S4, R1, R2 and H1, and it is characterized by $\tau_{sf} = 3.52 \epsilon r_0^{-2}$ (cf. Table 1 in the manuscript). The other adopted potential has the same equilibrium energy ϵ , but its equilibrium and cutoff distances are scaled by a factor 0.90 in simulation G1 and 0.95 in simulation G2. The potential well of the cross-terms between the two species is 0.9ϵ ; its equilibrium and cut-off lengths are

given by the average of the respective lengths of the two species. The grains are randomly distributed by means of Voronoi tessellation, the orientation of each of them determined by a random rotation.

What is observed is that the steady-state roughness appears to oscillate around larger values than the other cases. This is explainable with the competition between the embrittlement of the scaled down species and its increased toughness. A new preferential rupture is introduced (with respect to the homogeneous case): the material prefers to break within the grains made of the less tough potential or along the grain boundaries. Remarkably, the final surfaces still appear self-affine with Hurst exponent in the same range as for the other simulations. Again, we thank the reviewer for suggesting this new case study as it strengthens the message of the paper.

A paragraph has been added in the main manuscript where both the results obtained from the new simulations with grain boundaries and the effects of other micro-structures are discussed. The paragraph is the following:

‘The case of heterogeneous materials with grain boundaries (simulations G1 and G2) yields further observations. The inclusion of two different species and the presence of grains, instead of a perfect lattice, provide favourable crack propagation directions that are not present in the homogeneous cases. Cracks propagate more easily along the grain boundaries—where the mismatch of the lattices reduces the overall bond strength between the atoms at the interface—and within the bulk of the least tough species. Additionally, cracks are more likely to stop propagating when the tip meets the tougher material (cf. Supplementary Video 1). Furthermore, the heterogeneous grains modelled in simulations G1 and G2 can be seen as inclusions of one of the two species in a matrix made of the other one, in the extreme case where the two species have the same phase fraction. The same mechanisms affecting the crack path (because of the lattice mismatch and/or different material properties) are thus expected to occur—although more localized—when the inclusions’ phase fraction is smaller than the matrix phase fraction. As the latter is a case in between the homogeneous and heterogeneous bulk cases investigated here (i.e. simulations ‘R’ and ‘G’ respectively), the findings of this work suggest that a similar self-affine surface morphology should be recovered. Real materials might also contain pre-existing dislocations or point defects, which are not considered in the present set of simulations. In these cases, we would expect the defects to affect the plastic response of the bulk and, thus, its critical length scale and the debris formation process. Moreover, while we restricted ourselves to one possible crystal structure, materials can also be amorphous. Assuming homogeneous materials, we would then expect the crack path to have no favourable direction because of the isotropic structure, contrary to the hexagonal lattice modelled here. Finally, surfaces undergoing continuous reworking are also known to exhibit hardening, which most likely leads to a change of the critical length scale over time, affecting in turn the wear rate.’

5. The authors do discuss the fact that they made the choice of restricting the simulations to 2D systems in order to carry the simulations out to long time scales. This choice is understandable, but further discussion of the limitations of this are needed. The authors discuss possible consequences on page 6, lines 347-357. They point out that that “In real 3D tribosystems full adhesion might not take place.” The referee does not understand the point being made here; please clarify. The authors then state, “and the debris particles are not forced to pass over the same track again and again, possibly affecting the process as described above” (referring to the evacuation of debris). Even in 3D, there would not be any evacuation if two semi-infinite plane surfaces were simulated. However the referee agrees that particles would find locations where little to no contact occurs in 3D, which is not possible in 2D. The authors should discuss the limitations imposed by the 2D constraint in somewhat more depth. Their only statement is “Despite these differences, which might result in a slow down of the roughness evolution process, we argue that the rationale put forward should be recovered in 3D systems too, as the aforementioned evidence from experiments and field observation of faults suggests”. The referee agrees

a slow down of wear may occur for 3D vs. 2D simulations, but what other differences could be expected, including those resulting from the different nature of contact stresses for 3D vs. 2D systems? Strengthening this discussion further would be helpful. Could the authors run a short 3D simulation to at least compare what happens at the early time scales to see if there is some degree of commonality between 2D and 3D? That would strengthen the arguments of applicability for this paper.

We have conducted preliminary simulations¹ showing the existence of a critical length scale for the ductile-to-brittle transition also in 3D. Unfortunately we are unable to provide simulation data for roughness evolution in 3D because the computational cost of these simulations is out of reach for the clusters that we have access to at EPFL. We are hopeful that we will acquire such data in the future as we have been granted computing time on Swiss HPC clusters. In particular we plan to study the anisotropy of the surfaces by measuring the Hurst exponent parallel and perpendicular to the sliding direction. For the moment we restrict the paper to a simple discussion of what we expect to be relevant differences between 2D and 3D simulations. The updated paragraph is the following:

‘It should be also noted that our simulations are restricted to 2D systems by the large computational demands of long-timescale simulations in 3D. One relevant difference between 2D and 3D systems is that in the first case, the lattices of the two bodies are more likely to locally match, allowing full, bulk-like adhesion to develop. Furthermore, in 3D the debris particle could roll at some angle with respect to the sliding direction of the top body, as a consequence of local peaks and valleys due to the roughness along the additional dimension. It is also known that the steady-state roughness normal to the sliding direction is different than that parallel to the sliding direction², which cannot be captured by 2D simulations. Also, in 3D the debris particles are not forced to pass over the same track again and again. Despite these differences, which might result in a slowdown of the roughness evolution process, we believe that the same mechanisms should be recovered in 3D systems too, as the aforementioned evidence from experiments and field observation of faults suggests. Short timescale 3D simulations have also confirmed 2D observations for the early stages of the sliding process¹. Further investigations are necessary in any case to address the effects of 3D geometry and of the more complex micro-structures.’

6. It would further strengthen the paper if the authors could consider a finite contact geometry, e.g. a ball on a flat, where debris evacuation can occur. If similar trends are seen, then it would further strengthen the general applicability of the paper.

Although the point made by the reviewer is fair and the obtained insights would be of interest, we believe that carefully setting up, running and analyzing such a study would require a much longer time frame than what is possible within the current review process.

7. A minor point: in the conclusion, the authors state that their observation of a change in the wear rate contradicts Archard’s law and argue that this could be resolved if evacuation of debris were included. However, in macroscopic tests, changes in wear rate are often seen – high wear occurs during the ‘run in’ phase in the first few tens-100’s of cycles of most lab tests for a wide range of contact conditions. A steady wear rate following Archard’s law is often applied after this first transient phase. The fact that the authors see a change in behavior in the early stages of their simulation is consistent with many observations of wear.

The reviewer is of course correct. Nonetheless, in Archard’s model, the contact configuration is of the two-body type: he defined the wear coefficient itself as the probability of an asperity to break, once it comes in contact with another asperity. What we wanted to highlight is how a transition to a predominantly three-body contact configuration would change this concept and

might be responsible for a lower wear rate because of a lubricating effect. We agree that this was not clear in the previous version and have made this point clearer in the updated manuscript.

In summary, while this work considers a much longer time scale than any other wear simulations with this level of resolution, and even though a number of different conditions are varied, the other limitations of the work constrain the general applicability of the very interesting results. The referee is optimistic that the mechanism of self-affine surface topography generation the authors uncover would persist and be found in further simulations and thus a set of results that show this would firmly justify the paper's acceptance for publication in Nature. As it stands, the paper is still a very strong and innovative contribution, and at the very least is worth publication in Nature Materials.

3 Reviewer 3

Milanese and co-workers study the evolution of surface roughness during wear in a specific coarse-grained particle model developed by a subset of the authors in previous publications (Nature Comm. 7, 11816 (2016); PNAS 114, 7935 (2017); Phys. Rev. Lett. 120, 186105 (2018)). The calculations are identical to these previous works but the interpretation of the data is new. The authors claim that the surfaces that develop in their model are self-affine with a Hurst exponent of roughly 0.7, in agreement with many experimental findings, but I have doubts regarding this interpretation. While the overall subject matter is interesting, based on the model and interpretation of the outcome I do not recommend publication of this manuscript in Nature Communications. More detailed criticism follows.

3.1 Model

The “coarse-grained” model used by the authors cannot correspond to any physical system. It looks like an ad-hoc interatomic potential (that would then not be coarse-grained), but it is certainly not a systematic coarse-graining of some underlying atomic-scale model as is carried out in for example the polymer community. There are several inconsistencies that are (because of this) associated with the coarse-graining idea: Their model shows dislocation plasticity, but the scale of the particles is certainly larger than an atom (otherwise it wouldn't be coarse-grained). Dislocation as the carrier of plastic deformation here are not the mechanism to be expected at these coarse-grained scales. Similarly, the authors carry out their simulations at finite temperature but fluctuations decrease with coarse graining-level and may have no significance here. No systematic attempt to clarify lattice defects in the coarse-grained description and scaling of fluctuation with coarse-graining level. In addition, the model is two-dimensional and this may affect what the authors are seeing (e.g. the elastic response has a logarithmic, not a $1/r$, singularity; melting and therefore the effect of temperature is different in 2D and 3D), although my feeling is that this may not affect the overall results seen by the authors.

Numerical studies of model systems are useful, but those models should be clearly guided by physical reality and the farther the model is away from physical reality the less interesting I would judge the obtained outcome for a broad readership such as that of Nature Communications. I acknowledge that the fact that results obtained with this model have appeared in Nature Comm., PNAS and Phys. Rev. Lett. and that not all of my colleagues may share this view on the model.

We would first like to state that we do not make any claim that our model represents the most accurate possible discrete description of matter. Instead we aim to model a simple material, characterized by a set of discrete points, that permits the simultaneous description of plastic deformation and fracture phenomena at a desirable length scale. System responses beyond the elastic limit (i.e., plasticity and fracture) are controlled by interactions between those

points. For instance, the brittle/ductile response of the system, which is controlled through the competition between fracture energy and the energy associated with plastic slip, can be tuned by modifying the potential well, and the shape of the potential tail.

One could discuss if the term “coarse graining” is appropriate in the present case or not, but this is purely a matter of wording and does not concern the validity of the model and the significance of the results we obtain. Rather than arguing about the wording or the particulars of the model, we wish to point out that this is to the best of our knowledge the first model that even allows the observation of the phenomena described in the manuscript. Its value is in giving first insights into a complex process, which is heavily simplified out of necessity. We argue that such simplification itself is valuable and unavoidable, since the goal is to understand the adhesive wear process on a fundamental level. Of course, future work should include more materials related details, but it will be aimless without the guidance from relatively cheap, simple models. This is evidenced by the lack of fracture and re-roughening processes in previous atomistic studies of sliding contact^{3;4;5;6;7;8;9}, which is in part due to the large computational demands of more realistic interatomic potentials. With the help of the present work, we now know better what to look for and hope to motivate further work in the area.

While our work will definitely not be the last word on modelling adhesive wear, we believe that the novelty of being able to model such a roughness evolution and the huge importance—for a diverse set of industries and natural processes—of understanding wear processes make this work interesting for a broad readership, including the Physics, Geophysics, and Engineering communities.

3.2 Roughness

The authors quantify surface roughness in their calculations. They find a power-spectral density and height-difference autocorrelation function indicative of self-affine scaling. Scaling exponent is not clear. The authors claim $H=0.7$ but $H=0.5$ would equally well be possible given the spread of their data. (E.g. S1b is very close to 0.5.) The only clear conclusion to be drawn from PSD and height correlation is $0.5 < H < 1.0$ which is in the range of what is found experimentally.

Figure R.1 shows the PSD per unit length for the surface S1b, representative of the lower bound of the range $H = 0.6$ – 0.8 found, and that the Reviewer argues could equally well be fitted by $H = 0.5$. The Figure also shows the guide-lines corresponding to values of the Hurst exponent of $H = 0.5$ and $H = 0.6$. It is quite clear that the best fit between the two is given by $H = 0.6$, corresponding to the range 0.6 – 0.8 , which is what we report in the manuscript. Similarly, it is evident from the guide-lines of Figure 2a and Supplementary Figure 1a in the manuscript that $H = 1.0$ is way too large for an upper limit.

The images shown Fig. 1h and 1l illustrate that their surfaces are very flat but exhibit a series of steps that, judging from Fig. 3, lead to an rms amplitude of about five steps and a plateau width of about 20 particles. In other words, the height distribution is hardly continuous or a Gaussian. I find it hard to call these discrete surfaces “self-affine”. If I scale them down (according to the procedure described by the authors) this reduces the plateau width and clearly leads to surfaces that looks different; the notion of self-affinity for these discrete systems studied by the authors would need either rougher surfaces or larger areas (i.e. a continuum-like height distribution).

Figure R.2 reports the height distribution for surfaces taken from two different simulations (S1 and S4) and whose PSD is reported in the manuscript. We have examined all the surfaces, and we observe in general that surface heights are Gaussian or nearly-Gaussian; we only found

Figure R.1: PSD per unit length Φ for surface S1b. Black solid guide-line corresponds to Hurst exponent $H = 0.6$ and black dotted guide-line corresponds to $H = 0.5$.

Figure R.2: Height distribution for some sample surfaces, whose PSD is reported in the manuscript: simulations S1 (first row of plots) and S4 (bottom row of plots). In the legends, “top” and “bottom” refer to the top and bottom surface of the simulation respectively. Solid black curves represent the best Gaussian fit.

few cases where the distribution departs from a Gaussian one.

However, we do not understand why the reviewer expects the height distribution to be Gaussian, as it is by no means a necessary nor sufficient condition for self-affinity. For example, analytical surface growth models such as the Edwards-Wilkinson and the Kardar-Parisi-Zhang ones result in non-Gaussian height distributions¹⁰. Experimentally, while it is true that heights can appear Gaussian-distributed in self-affine surfaces, self-affine surfaces with non-Gaussian height distribution and Gaussian distributed heights belonging to non-self-affine surfaces are also found. We refer the reviewer to the review work in Ref. 11, and in particular to Figures 7 and 8 for the case of self-affine surfaces with non-Gaussian height distribution and to Figures 13 and 14 for non-self-affine surfaces with Gaussian height distribution.

Further evidence of self-affine surfaces with non-Gaussian height distribution is found in faults¹². It is also long known that the nature of the height distribution depends on the surface formation process and several different distributions can be found^{13;14} or assumed^{15;16;17} — the Gaussian distribution of heights is more of a mathematically convenient assumption, at least in the case of manufactured surfaces¹⁷.

We have also conducted new simulations G1 and G2, which evolve into steady-state surfaces which appear on average rougher (in terms of root mean square of heights) than the other simulations (cf. Figure 1p, Supplementary Figure 7a, Supplementary Video 1). The analysis shows a self-affine behaviour for these new surfaces too and is consistent with the results of the other simulations.

It is also worth noting that the Fourier transform of a step gives something that looks like

Figure R.3: Synthetic single-step surface. Left: geometry of the surface. Right: Analytical and numerically estimated PSD for the single-step surface. The black guide-line corresponds to an ideal Hurst exponent $H = 0.5$.

Hurst exponent 0.5. The height correlation of a step shows something similar.

We do not agree that the Fourier transform of a step can be confused with a self-affine surface. The Fourier transform of a window function is a sinc function^{18;19}. The PSD for a single step (or, equivalently, a window function) is then directly proportional to the square of a sinc function, whose value is zero at given intervals. This can be seen, for example, for the profile in Figure R.3 (left). The profile heights $h(x)$ are non-zero at 20 points out of the 330 that discretize the surface. The discretization resembles the one adopted for the surfaces in the manuscript. The analytical expression of the surface is

$$h(x) = \begin{cases} 1/S & \text{if } |x| \leq (1 - S)/2 \\ 0 & \text{if } |x| > (1 - S)/2 \end{cases}$$

and its PSD is then given by

$$PSD_h(f) = |\hat{h}(x)|^2 = |\text{sinc}(fS)|^2 = \left| \frac{\sin(\pi fS)}{\pi fS} \right|^2$$

where the hat indicates the Fourier transform of the function and S is the arbitrarily chosen window width. In our case $S = 20/330$, to resemble both our discretization and the plateau width conjectured by the reviewer.

Figure R.3 (right) shows both the numerical estimate and the analytical expression of the PSD of the stepped surface $h(x)$. The two are indistinguishable and the zero values (to machine precision) are evident. Clearly the PSD of the stepped surface does not resemble the PSD of a self-affine surface.

I invite the author to generate a line profile with 5-10 random steps and compute PSD and height correlation. It looks almost exactly like what is presented in this paper. I therefore believe the "self-affine scaling" reported by the authors is a artifact of their analysis.

Figure R.4: Synthetic stepped surface with $n = 16$ steps.

We do not agree that the self-affine scaling we report is an artifact of our analysis. We show in the following that our surfaces display non constant variance and stationary increments, which are properties of self-affine surfaces²⁰. Surfaces that are not self-affine, like the stepped surfaces suggested by the reviewer, do not have these properties.

Following the reviewer’s suggestion, we built the stepped surface shown in Figure R.4, which has $n = 16$ random steps (more than the 5-10 suggested by the reviewer). The surface has horizontal length $L = 1$ and is discretized in $p_{max} = 2048$ points. The surface is horizontally subdivided in n parts of equal length p_{max}/n , within each part a random site is picked and a step of random height and random horizontal length is inserted. Elsewhere, the height is set to 0. This procedure is equivalent to defining the heights $h(x_i)$ at each discretized point $i = 1, 2, \dots, p_{max}$ as

$$h(x_i) = \begin{cases} h_j & \text{if } x_k < x_i < x_{k+m}, \quad j = 1, 2, \dots, n \\ 0 & \text{otherwise} \end{cases}$$

where $h_j \in \mathbb{R}$ is randomly drawn from a normal distribution with mean 0 and variance 1, $k = q \cdot p_{max}/n + s$ (with $q = 0, 1, \dots, n - 1$ and $s \in \mathbb{N}$ randomly drawn from a discrete uniform distribution in the interval $[0, p_{max}/n]$), and $m \in \mathbb{N}$ is randomly drawn from a discrete uniform distribution in the interval $[0, p_{max}/n - s]$. To investigate the effect of sample size in the analysis, the first surface created has been discretized in smaller samples with $p = 256; 512; 1024$, where p evenly spaced points and their height are considered (instead of all the p_{max} points).

Figure R.5 shows the numerically estimated PSD of the created surface for the different sample sizes (i.e. the discretization). The convolution of several sinc functions smooths out the zero values and may mislead the interpretation of such plots at first sight, but the oscillations in the values are much larger than ours (cf. Figure 2a and Supplementary Figure 1a of the manuscript).

In addition, further checks can be performed to tell apart a stepped surface from a self-affine one. A self-affine surface is in fact a Non Stationary Process with Stationary Increments (NSPSI)²⁰: the non stationarity of the process means that the variance of the process changes with the sample size, while the stationarity of the increments implies that their average value does not change with the sample size or number of trials.

In the case of a surface, the variance is given by the root mean square of heights and an increment is the difference in height between two different points $\Delta h = h(x_i + \delta x) - h(x_i)$. In

Figure R.5: PSD of the synthetic stepped surface for four different resolutions. Black solid guide-line corresponds to an hypothetical Hurst exponent $H = 0.5$.

the case of self-affine surfaces, it is known that the root mean square of heights is size-dependent and increases with the sample size^{20;21}, as roughness is present at all scales. This property is displayed by the surfaces we analyzed, as the larger sample shows a larger root mean square of heights than the smaller ones, displaying a scaling that is compatible with self-affine surfaces (cf. Figure E.8 of the manuscript). On the other hand, this is not the case for a stepped surface, as it only has roughness at a given scale. As a consequence its root mean square of heights does not change with the sample size: this is confirmed for the synthetic surface described above and displayed in Figure R.4, as it can be seen in Table R.1.

p (–)	root mean square of heights (r_0)
256	0.535
512	0.535
1024	0.539
2048	0.537

Table R.1: Root mean square of heights for different discretizations p of the surface depicted in Figure R.4.

Furthermore, the stationarity of the increments in our case can be seen in Figure 2b and Supplementary Figure 1b (manuscript), where for different trials and different system sizes Δh does not change. On the other hand, again, it does change for the synthetic stepped surface (Figure R.6).

Recapping. Surfaces that satisfy the aforementioned properties, namely non-constant variance and stationary increments, have been shown to be NSPSI and self-affine²⁰. As shown above, this is the case for the surfaces we analysed in the main manuscript, but not for a stepped surface. By comparing our profiles with synthetic non-self-affine stepped profiles as suggested by the reviewer, it is evident that the observed self-affine scaling is not an artifact of our analysis.

Figure R.6: Height-height correlation function of the synthetic stepped surface for four different resolutions.

3.3 Junction size

The authors refer to an energy-balance criterion similar to Griffith’s for fracture that I believe goes back to Rabinowicz when talking about wear particle creation. They express this criterion in terms of what they call the critical junction size. This “junction-size” d was introduced in an earlier paper (Nature Comm. 7, 11816 (2016)) but appears to be an ill-defined and rather arbitrary concept. If I look at Fig. E6 then it is not clear to me why the junction size is determined but the red lines indicated by the authors and not by the shortest paths across the particle which is clearly smaller. There appears to be no clear definition of this junction size.

The concept introduced in Nat. Comm. 2016 indeed refers to a ductile-to-brittle-transition governed by the length scale of a contact junction and is well-defined there. This transition also applies in the present, more “messy” case (the local loading conditions are more complex and the geometry evolves due to plastic deformation during sliding). The old Extended Figure 6 was meant to depict schematically the mechanism of particle growth, and it was linked to the supplementary discussion on particle growth. It was then assumed that the junction coincided with the line of contact between the rolling particle and the opposing surface, e.g. in the instants before the frame depicted in the old Extended Figure 6, the atoms above the solid red lines belonged to the rolling particle, and those below to the opposing surface.

We thank the reviewer for the remark, as we have realized that the old Extended Figure 6 and the supplementary discussion about particle growth were out of the main focus of the manuscript and thus confusing the reader: we have decided to remove both of them.

According to the authors, the deformation should be plastic below a critical d^ but this is inconsistent with their own simulations. Once a particle has formed, the junction size (the point of contact between particle and surface) is very small and the deformation should be plastic, but no such plastic deformation appears to be observed. However, the authors don’t really need to refer to the junction size model for interpretation - I’m not sure why it is in this paper at all.*

We do not agree with this comment. We *do* observe plastic deformation for smaller contacting junctions (Figure R.7) and plasticity occurs continually at the contact between debris particle and surface (Figure R.8). Sometimes, though, the contact between particle and surface becomes big enough to induce fracture and growth of the particle, as depicted in the old Extended

Figure R.7: Snapshots from simulation R3. a) Original geometry. b) Zoom in to the asperities that will first collide. c-e) Evolution of the asperity collision. The asperities both undergo large deformation upon contact, the contact area increasing over time (cf. panels c and d), but never reaching a large enough size to create a loose debris particle and asperity separation is of the ductile type (e). Later contact between other asperities (cf. Figure 1 in manuscript) will display a large enough junction size to create a loose debris particle. Observations are consistent with the existence of a critical length scale for the ductile-to-brittle transition in the wear mechanism²³.

Figure R.8: Snapshot from simulation R2 (left), zoom in to contact zone (center) and in to a single dislocation (right). Black dotted lines enclose dislocations at and below the contact between debris particle and surface.

Figure 6, which leads to re-roughening of the surface and growth of the particle.

We argue that the inclusion of the critical length scale for the ductile-to-brittle transition is essential to capture the evolution to the self-affine morphology. This is because both the brittle detachment and ductile deformation processes are central to the development of the final surface roughness. Brittleness and ductility are not defined by material properties only, but by the junction length scales too. Earlier MD simulations⁸ and the Nat. Comm. 2016 paper show that, upon sliding, purely plastic deformation asymptotically leads to flattening and (depending on the adhesion between the surfaces) cold welding. The mechanistic transition described by the critical length scale concept is essential to understand the processes and length scales involved in the formation of rough surfaces (and is made computationally feasible with the present model approach). Without such a transition, the material behaviour would be either completely plastic or completely brittle, which goes against experimental observations: Rabinowicz already observed (e.g. in his book²²) that there is a minimal wear particle size, which also depends on the material under investigation. This is consistent with the critical length scale concept, that we proposed²³. However, our concept differentiates itself from Rabinowicz's criterion in the sense that the latter merely describes the minimum particle size, but does not refer to the notion of surface roughness evolution. The present work is guided by this idea: The formation and evolution of a wear particle is intimately connected to the ductile-to-brittle transition and in turn also determines the evolution of the surface roughness.

3.4 Other points

** I don't think Da Vinci, Amonton and Coulomb were aware of the importance of surface roughness. This is typically attributed to Bowden & Tabor, but the earliest reports date to the end of the 19th beginning of the 20th century.*

From the cited papers it appears clear that Da Vinci and Coulomb were aware that the surface roughness had implications in tribological applications (see citations below), even if they clearly could not have the picture that is available to us today. On a second read of Ref. 24 it is less clear for Amonton and we thus removed his name from the text.

- From Ref. 24: *'Coulomb investigated the force of friction as function of many factors, which Gillmor summarizes in the following list: [...] 2. surface conditions (polished, rough).'*
- From Ref. 25: *'[Da Vinci] wrote '..... different bodies have different kinds of friction; because if there shall be two bodies with different surfaces, that is that one is delicate and smooth and well-greased or soaped, and it is moved upon a plane of a similar nature, it will move much more easily than one that has been roughened by the use of file 8 or rasp' (Forster II 87r, c. 1497).'*

** What is meant by "different universal dimensionalities" in the introduction? I suppose the authors mean Hurst exponents or fractal dimensions?*

Yes, we are referring to the Hurst exponents. We changed the expression to 'different Hurst exponents' to improve the comprehensibility, as that is actually how we address that concept in the rest of the manuscript.

** Model is called coarse-grained and initially then later referred to as atomistic.*

We called the simulations 'atomistic' to refer to the general numerical method. Following the reviewer's comment, we changed this to 'molecular dynamics simulations' as it is more generic.

** What does the Boltzmann constant k_B mean in the context of the coarse-grained model and what is its value, i.e. how is temperature measured? Did the authors defined some reference temperature, e.g. the melting temperature of the solid? (Melting in 2D is very different than in 3D so this may not be very meaningful.) See also my comment above concerning temperature and coarse-graining.*

We thank the reviewer for pointing this out. The definition of the Boltzmann constant remains unchanged from the usual definition. The reviewer is correct that in order to define a temperature unit as ε/k , k is only the Boltzmann constant if all degrees of freedom are explicitly simulated, i.e., the particles are not coarse-grained and correspond to atoms. Since we do not convert to an actual temperature in non-reduced units, this is only a notational issue, which we fixed in the updated manuscript by explicitly stating in the methods that the values for the temperature are given in equivalent kinetic energy. We do not consider a specific material, and thus did not calibrate melting points to experimental data.

** If the authors use a standard Langevin thermostat, then their friction force will be the damping force of the Langevin because the Langevin thermostat contains reference velocity which is zero by default. How did the author do the thermalization exactly?*

We use the 'compute group/group' command implemented in LAMMPS, in which the calculated force is 'the force on the compute group atoms due to pairwise interactions with atoms in the

Figure R.9: Measured tangential force in the bottom region for simulation G1, with applied Langevin thermostat (orange dots) and without (light blue dots).

specified group2¹. In our case, the first group contains the bottom atoms on which boundary conditions are applied (i.e. those whose velocity is fixed to zero and those in the bottom thermostatted region) and group2 contains all the atoms free of any applied boundary condition. As a result, the contributions due to the drag and random forces introduced by the Langevin thermostat are not included in the tangential force we reported. The estimated tangential force is thus the horizontal component of the conservative force transmitted by the atoms free of any imposed condition on those belonging to the bottom thermostat or to the bottom fixed region.

Nevertheless, to verify our setup, we re-ran part of simulation G1 without any thermostat and compared the tangential force with our previous simulations. The measurements are reported in Figure R.9 and it is clear that no effect due to the thermostat is present in our measurements. The two measurements are in fact indistinguishable for a significant part of the simulations, after which the system without thermostat starts to heat up and thus diverges from the trajectory of the thermostatted system.

** The authors say fractional Brownian motion has non-stationary increments. This is wrong. The increment process (fractional Gaussian noise) is stationary.*

We agree and thank the reviewer for catching this. The manuscript has been updated.

** What does the sentence "even if two different critical length scales coexist" mean?*

As the bulk is made of two different materials (in simulations H1 and, now, G1 and G2 too), ideally two different critical length scales are modelled at the same time. At this stage it is not known if this heterogeneity results in a globally different critical length scale for the whole heterogeneous material, if one of the two critical length scales prevails upon asperity collision, or if some other scenario takes place. We reformulated the manuscript to clarify this point.

** How did the authors extract the roughness? I guess it was mapped onto a grid (of which*

¹https://lammps.sandia.gov/doc/compute_group_group.html

spacing?) before carrying out the Fourier transform. How were the wear particles that are still attached to a surface at the end of the calculation removed?

A script was developed to detect and remove the fragment from LAMMPS output files. We expanded the *Data analysis* paragraph of the Methods section explaining the steps of the procedure, including also remarks from Reviewer 1.

* *FFT(z) should be specified explicitly because normalization factors can vary. Indeed FFT is an algorithm. I believe the authors mean the discrete Fourier transform.*

We thank the reviewer for the comment and we agree that we over-simplified this part of the Methods section. We have therefore expanded it, explaining how the discrete estimation relates to the continuous one and showing the adopted normalization. Because of this we changed the notation for the adopted PSD estimation and therefore updated the text and figure labels accordingly.

* *Fig. 3 should have a scale bar.*

In fact, Fig. 3 is a graph where all axes are labelled. In case the referee refers to a figure with a snapshot of the simulation, we would like to point out that Fig. 1 contains labels that indicate the lengths l_x and l_y , the values of which are given in Table 1 for all simulations.

* *As already stated above, it would be interesting to see height distributions.*

Height distributions for some sample surfaces are plotted in Figures R.2 and discussed in the Roughness section (Sec.3.2).

References

- [1] Aghababaei, R., Warner, D. H. & Molinari, J.-F. On the debris-level origins of adhesive wear. *Proc. Natl. Acad. Sci.* **114**, 7935–7940 (2017).
- [2] Sagy, A., Brodsky, E. E. & Axen, G. J. Evolution of fault-surface roughness with slip. *Geology.* **35**, 283–286 (2007).
- [3] Stoyanov, P., Romero, P. A., Merz, R., Kopnarski, M., Stricker, M., Stemmer, P., Dienwiebel, M. & Moseler, M. Nanoscale sliding friction phenomena at the interface of diamond-like carbon and tungsten. *Acta Materialia* **67**, 395–408 (2014).
- [4] Pastewka, L., Moser, S., Gumbsch, P. & Moseler, M. Anisotropic mechanical amorphization drives wear in diamond. *Nat. Mater.* **10**, 34–38 (2011).
- [5] Bouchet, M. D. B., Matta, C., Vacher, B., Le-Mogne, T., Martin, J., von Lautz, J., Ma, T., Pastewka, L., Otschik, J., Gumbsch, P. & Moseler, M. Energy filtering transmission electron microscopy and atomistic simulations of tribo-induced hybridization change of nanocrystalline diamond coating. *Carbon* **87**, 317–329 (2015).
- [6] Zhong, J., Shakiba, R. & Adams, J. B. Molecular dynamics simulation of severe adhesive wear on a rough aluminum substrate. *J. Phys. D: Appl. Phys.* **46**, 055307 (2013).
- [7] Sha, Z.-D., Sorkin, V., Branicio, P. S., Pei, Q.-X., Zhang, Y.-W. & Srolovitz, D. J. Large-scale molecular dynamics simulations of wear in diamond-like carbon at the nanoscale. *Appl. Phys. Lett.* **103**, 073118 (2013).

- [8] Spijker, P., Anciaux, G. & Molinari, J.-F. Dry sliding contact between rough surfaces at the atomistic scale. *Tribol. Lett.* **44**, 279 (2011).
- [9] Sorensen, M., Jacobsen, K. W. & Stoltze, P. Simulations of atomic-scale sliding friction. *Phys. Rev. B* **53**, 2101–2113 (1996).
- [10] Zhao, Y.-P., Wang, G.-C. & Lu, T.-M. Diffraction from non-gaussian rough surfaces. *Phys. Rev. B* **55**, 13938 (1997).
- [11] Persson, B., Albohr, O., Tartaglino, U., Volokitin, A. & Tosatti, E. On the nature of surface roughness with application to contact mechanics, sealing, rubber friction and adhesion. *J. Phys.: Condens. Matter* **17**, R1 (2004).
- [12] Renard, F., Voisin, C., Marsan, D. & Schmittbuhl, J. High resolution 3d laser scanner measurements of a strike-slip fault quantify its morphological anisotropy at all scales. *Geophys. Res. Lett.* **33** (2006).
- [13] Williamson, J., Pullen, J., Hunt, R. & Leonard, D. The shape of solid surfaces. *Surf. Mech. ASME, New York* 24–35 (1969).
- [14] Peklenik, J. Paper 24: New developments in surface characterization and measurements by means of random process analysis. In *Proceedings of the Institution of Mechanical Engineers, Conference Proceedings*, vol. 182, 108–126 (SAGE Publications Sage UK: London, England, (1967).
- [15] Yu, N. & Polycarpou, A. A. Contact of rough surfaces with asymmetric distribution of asperity heights. *Transactions ASME-F-Journal Tribol.* **124**, 367–376 (2002).
- [16] Tayebi, N. & Polycarpou, A. A. Modeling the effect of skewness and kurtosis on the static friction coefficient of rough surfaces. *Tribol. international* **37**, 491–505 (2004).
- [17] Thomas, T. R. *Rough surfaces*, vol. 2 (Imperial College Press London, 1999).
- [18] Press, W. H., Teukolsky, S. A., Vetterling, W. T. & Flannery, B. P. *Numerical recipes 3rd edition: The art of scientific computing* (Cambridge university press, 2007).
- [19] VanderPlas, J. T. Understanding the Lomb–Scargle periodogram. *The Astrophys. J. Suppl. Ser.* **236**, 16 (2018).
- [20] Ganti, S. & Bhushan, B. Generalized fractal analysis and its applications to engineering surfaces. *Wear* **180**, 17–34 (1995).
- [21] Majumdar, A. & Tien, C. Fractal characterization and simulation of rough surfaces. *Wear* **136**, 313–327 (1990).
- [22] Rabinowicz, E. *Friction and wear of materials* (Wiley, New York, 1995).
- [23] Aghababaei, R., Warner, D. H. & Molinari, J.-F. Critical length scale controls adhesive wear mechanisms. *Nat. Commun.* **7** (2016).
- [24] Popova, E. & Popov, V. L. The research works of Coulomb and Amontons and generalized laws of friction. *Friction.* **3**, 183–190 (2015).
- [25] Hutchings, I. M. Leonardo da Vinci’s studies of friction. *Wear* **360**, 51–66 (2016).

Reviewers' comments:

Reviewer #1 (Remarks to the Author):

The authors have addressed in detail all the concerns raised in the refereeing process, or have provided convincing scientific arguments in support of their views. The detailed discussion contained in file 172000_1_rebuttal_3318409_pg4qvx.pdf is an interesting paper in itself.

As it stands now, the manuscript addresses an important scientific question and reaches the quality standards for publication in Nature Communications. My recommendation is to publish it as it stands.

Regrettably I could not view the file 172000_1_additional_review_material_3324674_pgc38g.pdf which relies on some nonstandard extension only compatible with one proprietary pdf viewer. I suspect that, whatever content is there, it would not change my recommendation.

Reviewer #2 (Remarks to the Author):

The authors have substantially improved the manuscript (which was already very strong with intriguing results on an important and general problem of broad interest). Their new results, particularly the new studies of polycrystalline systems and surfaces with a different Hurst exponent, significantly strengthen the points they make. I fully support publication.

I do strongly recommend the authors cite recent work that supports the results of this manuscript quite nicely, where a Hurst exponent of 0.75 ± 0.05 (!) is reported for rock surfaces down to the nanoscale, and importantly, contains a mechanics-based explanation for the observation:

Thom, C., Brodsky, E., Carpick, R., Pharr, G., Oliver, W. & 573
Goldsby, D. Nanoscale roughness of natural fault surfaces con- 574
trolled by scale-dependent yield strength. Geophys. Res. Lett. 575
44, (2017).

This paper was cited in the original submission of the manuscript. Some brief discussion on this paper, particularly since it appears to be the first to propose that mechanical behavior underlies the persistent 0.7 Hurst exponent, is warranted.

Some minor grammar errors need addressing through proofreading.

Overall, this is a strong and novel contribution that advances the field and will inspire new efforts.

Reviewer #3 (Remarks to the Author):

Milanese et al. have responded to my comments and some have been satisfactorily resolved. I refer below only to the comments where concerns remain. My main concern, namely the data does not support the authors' conclusions on self-affinity, remain unresolved.

Model

The authors respond: "One could discuss if the term "coarse graining" is appropriate in the present case or not, but this is purely a matter of wording and does not concern the validity of the model and the significance of the results we obtain."

I find this a rather awkward interpretation of a model. A model is supposed to describe reality and it is therefore not purely a matter of wording. As already indicate in my initial response I will not rest on this issue but strongly urge the authors to find a more first-principles justification for the model they employ.

Roughness

Self-affinity of surfaces:

1) The authors are correct that surfaces can be non-Gaussian - the key point is having a continuous height distribution (which in many cases can be Gaussian). The authors now present height distribution but I cannot understand how these were obtained. Let's look at surface S1: Supposedly the surface shown in Fig. 1d and the top row of R2. The picture is a bit blurry but it looks like in the top surface of Fig. 1d, maximum and minimum points on the surface are ~ 10 layers of the lattice apart. The corresponding distribution function can therefore not be continuous but must consist of ~ 10 discrete points; the system size is too small to allow for that conclusion and this is precisely the point I made in my earlier report.

The spacing between each layer is $\sqrt{3}/2 r_0 \sim 0.9 r_0$, hence the spread of the distribution function should be around $9 r_0$ and it should only have values at integer multiples of $0.9 r_0$. However, the distribution in Fig. R. 2 goes from $-12 r_0$ to $12 r_0$ in steps of what looks $< 0.9 r_0$. Was the distribution sampled including thermal fluctuations? Note that even for a flat surface thermal fluctuations will lead to a Gaussian looking height distribution, but the rms width should be smaller than r_0 . I simple cannot understand how the distribution of Fig. R. 2 can correspond to what is shown in Fig. 1 unless Fig. 1 does not show the full surface.

2) The visual test of Fig. R. 1 on the value of the exponent does not clearly show that 0.6 is the Hurst exponent, and the authors should probably quantify the (statistical) error on those fits. Additional (systematic) errors on fitting exponents can be large (e.g. Schmittbuhl, PRE 51, 131 (1995)) and the spread at large q in the data allows for both fits. The authors could plot $q^H * PSD$ for the two values to more clearly indicate the spread of the data.

3) PSD of a step: The artificial surface created by a random process model involving individual steps of the authors cannot correspond to their simulation, since

- a) the step height does not vary continuously in the simulation (but it does in their random process model) and
- b) the step position is not chose randomly by at equidistant points (but the position is not equidistant in their particle model) and
- c) the scaling analysis is not carried out at constant step density but at constant total number of steps.

It is therefore not a model for a random surface with random (uncorrelated) positions of the steps and I would not expect the surface to be self-affine. Random positions of steps must lead to 0.5 in the PSD since this is nothing else than a random walk and the rms vs system size must then scale with the same Hurst exponent. Yet, the authors' surface shows $H \sim 0.5$, emphasizing my point that this is an artifact that can be obtained by various routes.

The authors then say: "This property is displayed by the surfaces we analyzed, as the larger sample shows a larger root mean square of heights than the smaller ones, displaying a scaling that

is compatible with self-affine surfaces (cf. Figure E.8 of the manuscript)." I agree that this type of scaling would be indicative of self-affinity, but cannot find Figure E.8 to confirm what it shows.

What I am saying, in summary, is that the data (surfaces shown in Fig. 1) does not support the conclusions (surfaces are self-affine) and this concern has not been resolved. The system size studied by the authors is too small to conclude anything on the fractal character of the surfaces. Additionally, the stepped, random process surfaces the authors present in their response are erroneously created and cannot be self-affine.

3) Role of the "critical length scale": The authors argue "that the inclusion of the critical length scale for the ductile-to-brittle transition is essential to capture the evolution to the self-affine morphology" but this length scale does not appear to affect the outcome of their simulation. I agree that a ductile-to-brittle transition is necessary to explain the authors results. However, a length-dependent ductile-to-brittle transition is also documented elsewhere, e.g. in the indentation literature; this is not a new concept.

4) I find the name "molecular dynamics simulations" unfortunate for the authors model. As argued in my earlier report, there is no molecular scale in this model. "Discrete element method" or "mesoparticle method" would capture the essence better.

5) Langevin thermostat: By point was not on how the force is computed by that the Langevin damping affects the simulated dissipation. However, the NVE test is suitable to show that the influence of thermostating is negligible.

6) The authors still do not explain how the surface is extracted in the updated methods section. What is the value of Δx ? Is this the maximum atom in bins of size Δx ? Can more than one surface atom be in one bin?

We are thankful to reviewers 1 and 2 for supporting our work, and to reviewer 3 for providing additional comments and questions that have helped strengthen the message of our manuscript.

1 Reviewer 1

The authors have addressed in detail all the concerns raised in the refereeing process, or have provided convincing scientific arguments in support of their views. The detailed discussion contained in file 172000_1_rebuttal_3318409_pg4qvx.pdf is an interesting paper in itself.

As it stands now, the manuscript addresses an important scientific question and reaches the quality standards for publication in Nature Communications. My recommendation is to publish it as it stands.

Regrettably I could not view the file 172000_1_additional_review_material_3324674_pgc38g.pdf which relies on some nonstandard extension only compatible with one proprietary pdf viewer. I suspect that, whatever content is there, it would not change my recommendation.

We thank the reviewer for the kind words.

The file that the reviewer could not read is probably the “editorial policy checklist”, whose format is imposed by Springer Nature, but which is just a declaration of compliance with journal policies.

2 Reviewer 2

The authors have substantially improved the manuscript (which was already very strong with intriguing results on an important and general problem of broad interest). Their new results, particularly the new studies of polycrystalline systems and surfaces with a different Hurst exponent, significantly strengthen the points they make. I fully support publication.

I do strongly recommend the authors cite recent work that supports the results of this manuscript quite nicely, where a Hurst exponent of 0.75 ± 0.05 (!) is reported for rock surfaces down to the nanoscale, and importantly, contains a mechanics-based explanation for the observation:

Thom, C., Brodsky, E., Carpick, R., Pharr, G., Oliver, W., Goldsby, D. Nanoscale roughness of natural fault surfaces controlled by scale-dependent yield strength. Geophys. Res. Lett. 44, (2017).

This paper was cited in the original submission of the manuscript. Some brief discussion on this paper, particularly since it appears to be the first to propose that mechanical behavior underlies the persistent 0.7 Hurst exponent, is warranted.

Some minor grammar errors need addressing through proofreading.

Overall, this is a strong and novel contribution that advances the field and will inspire new efforts.

We thank the reviewer for the comment. This work was cited since the first submission but we agree that it is worth discussing it in more detail and we thus added the following sentence in the manuscript:

“The inclusion of a scale-dependent material strength is likely another fundamental ingredient needed in theoretical models to capture a persistent Hurst exponent (i.e. $H > 0.5$, Thom et al. (2017)). There is in fact evidence (Thom et al. (2017)) that mechanical behaviour underlies a Hurst exponent $H = 0.75 \pm 0.05$ in rocks at the nanoscale.”

3 Reviewer 3

Milanese et al. have responded to my comments and some have been satisfactorily resolved. I refer below only to the comments where concerns remain. My main concern, namely the data does not support the authors' conclusions on self-affinity, remain unresolved.

3.1 Model

The authors respond: "One could discuss if the term "coarse graining" is appropriate in the present case or not, but this is purely a matter of wording and does not concern the validity of the model and the significance of the results we obtain."

I find this a rather awkward interpretation of a model. A model is supposed to describe reality and it is therefore not purely a matter of wording. As already indicate in my initial response I will not rest on this issue but strongly urge the authors to find a more first-principles justification for the model they employ.

The first-principles justification for our model is that it includes the necessary ingredients of the adhesive wear process, namely adhesion, plastic slip, and fracture. These are the expected phenomena in the setup we are investigating. The pair potentials we use describe the discrete nature of these phenomena. The barriers for plasticity and fracture can be tuned in order to enable us to investigate different regimes in the wear process for the first time; in the present case we study a material which exhibits both of them at the relevant length scale of our simulation. To settle this point of disagreement with the reviewer, we have decided to remove the mention of "coarse-grained" models at the two locations where it appeared in the text, because it suffices to say that we are conducting molecular dynamics simulations with simple pair potentials.

3.2 Self-affinity of surfaces

1) The authors are correct that surfaces can be non-Gaussian - the key point is having a continuous height distribution (which in many cases can be Gaussian). The authors now present height distribution but I cannot understand how these were obtained. Let's look at surface S1: Supposedly the surface shown in Fig. 1d and the top row of R2. The picture is a bit blurry but it looks like in the top surface of Fig. 1d, maximum and minimum points on the surface are ~ 10 layers of the lattice apart. The corresponding distribution function can therefore not be continuous but must consist of ~ 10 discrete points; the system size is too small to allow for that conclusion and this is precisely the point I made in my earlier report.

The spacing between each layer is $\sqrt{3}/2 r_0 \sim 0.9 r_0$, hence the spread of the distribution function should be around $9 r_0$ and it should only have values at integer multiples of $0.9 r_0$. However, the distribution in Fig. R. 2 goes from $-12 r_0$ to $12 r_0$ in steps of what looks $< 0.9 r_0$. Was the distribution sampled including thermal fluctuations? Note that even for a flat surface thermal fluctuations will lead to a Gaussian looking height distribution, but the rms width should be smaller than r_0 . I simple cannot understand how the distribution of Fig. R. 2 can correspond to what is shown in Fig. 1 unless Fig. 1 does not show the full surface.

We thank the reviewer for bringing back this point. We discuss below that the lattice planes are not aligned in all parts of the sample, thus the height distributions of our surfaces are not confined to integer multiples of $\sqrt{3}/2 r_0$ (as shown in Figures R.1 and R.2). Nonetheless, we disagree that the continuity of the height distribution is a requirement for a surface to be self-affine.

We wish to emphasize that the process of roughness creation during adhesive wear is not an uncorrelated random process. A debris particle that was formed thanks to a contact junction reaching a critical size is constantly working the surface and subsurface locally and with an im-

Figure R.1: Height distributions for surfaces of simulation S1 at sliding distance $29950 r_0$, as in Figure 1d of the manuscript, after minimization to avoid effects due to temperature fluctuations. Heights are distributed in bins of size $0.1 \cdot \sqrt{3}/2 r_0$, corresponding to one tenth of the lattice spacing in the y direction. Values in labels along the x axis are in multiples of $2 \cdot \sqrt{3}/2 r_0$.

posed rolling direction. In the process dislocations are nucleated and stored, and in the presence of a micro-structure, grain boundaries slide and grains rotate. These processes all interact with each other.

Here, Figure R.1 reports the height distribution for the top and bottom surfaces of the single frame shown in Figure 1d, after the system is relaxed at zero temperature (the debris particle being already identified and removed as described in the Methods). The minimum height y_{min} of each set of particles is subtracted from the y coordinate of each particle and the particles are then distributed in bins of size $0.1 \cdot \sqrt{3}/2 r_0$ (i.e. one tenth of the lattice spacing in the y direction). The Figure clearly shows that the atoms are not confined to integer multiples of $\sqrt{3}/2 r_0$.

Even more evident is the case with grain boundaries that we introduced in the second submission thanks to the remarks of reviewer 2. Figure R.2 reports the height distribution for the case of simulation G1 at the timestep depicted in Figure 1p. As in the previous case, the system is minimized and heights $y - y_{min}$ are distributed in bins of size $0.1 \cdot \sqrt{3}/2 r_0$. The spread of heights is clearly visible.

In both cases, the lattice planes are not aligned in all parts of the sample, both because of local deformation (when the debris particle is in contact with the surface) and because of permanent deformation due to dislocations (see Figure R.8 of the previous rebuttal letter). Furthermore, in the case of grain boundaries each grain has a different lattice rotation (which can change during the wear process) and atoms are never exactly positioned at coordinates which are multiples of $\sqrt{3}/2 r_0$.

Note also that, as stated in the main manuscript, for the top and bottom surfaces of each simulation, the PSD is averaged over several time frames (as recommended in the literature; see for instance the reference textbook Barabási and Stanley (1995)). This is done by computing the PSD of each surface separately at each time frame, and then averaging the PSDs. Thus, we also showed an average height distribution in the last rebuttal letter.

We now wish to go back to the self-affinity of interfaces with regard to discrete versus continuous systems. Examples abound in the literature in which discrete systems are used to study fractal surfaces. Barabási and Stanley (1995) provide a comprehensive collection of discrete surface growth models, such as (but not limited to) a random deposition model with surface relaxation, the directed polymer model, and several discrete models put forward to investigate molecular beam epitaxy. For the latter, where the scale is atomistic, numerical simulations on system sizes L in the range $[10; 450]$ (in number of discrete points) are reported. More recently,

Figure R.2: Height distributions for surfaces of simulation G1 at sliding distance $29750 r_0$, as in Figure 1p of the manuscript, after minimization to avoid effects due to temperature fluctuations. Heights are distributed in bins of size $0.1 \cdot \sqrt{3}/2 r_0$, corresponding to one tenth of the lattice spacing in the y direction. Values in labels along the x axis are in multiples of $10 \cdot \sqrt{3}/2 r_0$. Subticks are $\sqrt{3}/2 r_0$ apart.

Gjerdén et al. (2013) investigated the roughness of a crack front by means of an extended fiber bundle model, with system sizes L in the interval $[64; 512]$. Finally, Luan and Robbins (2009) model rough surfaces at the atomistic scale in a similar fashion, i.e. by arranging the atoms in a triangular lattice and successively removing those whose height y is larger than the height $h(x)$ prescribed by a self-affine surface: the resulting surface atoms all lie on crystal planes.

2) The visual test of Fig. R. 1 on the value of the exponent does not clearly show that 0.6 is the Hurst exponent, and the authors should probably quantify the (statistical) error on those fits. Additional (systematic) errors on fitting exponents can be large (e.g. Schmittbuhl, PRE 51, 131 (1995)) and the spread at large q in the data allows for both fits. The authors could plot $q^H \cdot PSD$ for the two values to more clearly indicate the spread of the data.

The spread at large wavevectors is not meaningful and was excluded from the fitting, as it is subject to artifacts due to being close to the inter-particle spacing r_0 , an unclear definition of surface at that scale, surface artifacts such as small overhangs, and thermal fluctuations. As such, it should not be used to judge the fit (cf. Figure 11 in Schmittbuhl et al. (1995)).

Schmittbuhl et al. (1995) also strongly recommend to adopt different methods and different sample sizes, which we followed by investigating two different sample sizes (see below) and analyzing all the surfaces using both the PSD and the height-height correlation function, as recommended by Barabási and Stanley (1995) as well.

Schmittbuhl et al. (1995) do not consider the statistical error in their fit, but the systematic error only. This was estimated as the difference between the “input” and “output” values of H . In our case and in most measurements, the “input” value of H is unknown. Schmittbuhl et al. (1995) find that the power spectrum method (that we also use) has the tendency to underestimate the Hurst exponent: this reduces even more the possibility that our fit ($H = 0.6$ for surface S1b) hides a lower exponent (e.g. $H = 0.5$ as suggested by the reviewer). At most, as suggested by Table I of Schmittbuhl et al. (1995), we may be underestimating our Hurst exponent by 0.1. For the case of surface S1b it could then be $H = 0.7$.

Applying this systematic error to the range of Hurst exponents for all the surfaces indicated in the main manuscript, $H = 0.6 - 0.8$, it would become at most $H = 0.7 - 0.9$. We thank the reviewer for this comment, and we have expanded the Methods section of the main manuscript

to address this.

Concerning the statistical error of the fits, the error bars often reported in the literature refer to the error of the linear fit on the log-log plot, and in our case this is in the order of ± 0.05 . As we do not claim an exact value of the Hurst exponent, rather an interval of values (i.e. $H = 0.6 - 0.8$), the claimed range already includes the statistical error.

To confirm the goodness of the fit, we follow here the reviewer’s advice and Figure R.3 shows the error $E(H, q) = \Phi_q / (b \cdot q^{-\alpha_H}) - 1$ as a function of the wavevector q , where Φ_q is the PSD per unit length extracted from the simulation data, $\alpha_H = 2 \cdot H + 1$ is the power law exponent, and b is the intercept in the log-log plot determined by the linear fit for a given value of α_H . Subscripts indicate the Φ_q and α_H are functions of q and H respectively. We considered the bottom surface of simulation S1 (the same considered in the previous rebuttal letter). Three different values of H have been tested: $H = [0.5, 0.6, 0.7]$. All the fits have been performed for values of the wavevector up to $q = 1.57 r_0^{-1}$ (i.e. $\lambda = 4.00 r_0$). It can be clearly seen that $E(H = 0.6, q)$ is the error that is the closest to the zero error line, while $E(H = 0.5, q)$ and $E(H = 0.7, q)$ tend to respectively overestimate and underestimate the value of $\Phi(q)$ at small wavevectors (and vice versa for high q). This is consistent with the fact that to a change in the Hurst exponent corresponds a change in the slope in the log-log plot.

3) PSD of a step: The artificial surface created by a random process model involving individual steps of the authors cannot correspond to their simulation, since

a) the step height does not vary continuously in the simulation (but it does in their random process model) and b) the step position is not chose randomly by at equidistant points (but the position is not equidistant in their particle model) and c) the scaling analysis is not carried out at constant step density but at constant total number of steps.

It is therefore not a model for a random surface with random (uncorrelated) positions of the steps and I would not expect the surface to be self-affine. Random positions of steps must lead to 0.5 in the PSD since this is nothing else than a random walk and the rms vs system size must then scale with the same Hurst exponent. Yet, the authors’ surface shows $H = 0.5$, emphasizing my point that this is an artifact that can be obtained by various routes.

We believe that we have already addressed this point above. The fact that a random walk would lead to a PSD that looks distributed with a Hurst exponent $H = 0.5$ just cannot relate to our findings, as our data shows $H = 0.6 - 0.8$, well above the random walk exponent. This is a clear hint that Gaussian randomness alone cannot explain the value of the Hurst exponent of our surfaces, and is consistent with several other findings reported in the literature (see references in main text). Considering a systematic error of at most 0.1 as found by Schmittbuhl et al. (1995) pushes our interval even further from $H = 0.5$. Also, as shown above, our height distributions do not correspond to a few isolated steps.

Furthermore, the process of adhesive wear is not a Gaussian random process (random walk), and the material is constantly being worked by the dragged debris particle. So we do not understand why the reviewer expects us to find $H = 0.5$, which would correspond to a random walk. Reviewer 2 points to the contrary to experimental evidence of $H = 0.75$ due to dry sliding wear (Thom et al. (2017)).

If the reviewer is referring to the synthetic surfaces of the first rebuttal letter instead, they do not show $H = 0.5$, as explained (cf. Figures R.3 and R.5 of the previous rebuttal).

Finally, it is not clear to us what the reviewer means when he/she writes: “It is therefore not a model for a random surface with random (uncorrelated) positions of the steps and I would not expect the surface to be self-affine”. The reviewer seems to imply that a model with random uncorrelated heights is self-affine. This is wrong, as clearly shown by the random deposition model, in both its discrete and continuum versions, where each site’s height is independent of

Figure R.3: Error of the PSD per unit length Φ for bottom surface of simulation S1b with different Hurst exponent $H = [0.5, 0.6, 0.7]$. $H = 0.6$ corresponds to the best fit. Black solid guide-line corresponds to an error equal to zero. Black dotted guide-line shows the maximum frequency adopted for the linear fit.

the others — the correlation length is then zero, no Hurst exponent can be defined and the surface is *not* self-affine (Barabási and Stanley (1995)).

The authors then say: "This property is displayed by the surfaces we analyzed, as the larger sample shows a larger root mean square of heights than the smaller ones, displaying a scaling that is compatible with self-affine surfaces (cf. Figure E.8 of the manuscript)." I agree that this type of scaling would be indicative of self-affinity, but cannot find Figure E.8 to confirm what it shows.

The figure has been there since the first submission, where it was numbered "Figure E.8". Since the second submission, it is numbered "Supplementary Figure 2", and it is referred to in the main text when the size effect is discussed (lines 213-216 in the second submission). We are happy that the reviewer supports our argument that such scaling of r.m.s. of heights with system size is indicative of self-affinity.

What I am saying, in summary, is that the data (surfaces shown in Fig. 1) does not support the conclusions (surfaces are self-affine) and this concern has not been resolved. The system size studied by the authors is too small to conclude anything on the fractal character of the surfaces. Additionally, the stepped, random process surfaces the authors present in their response are erroneously created and cannot be self-affine.

We believe that our previous and above arguments should address all these concerns. In particular, discrete systems and relatively small system sizes have been investigated before, with no loss of generality, both experimentally (e.g. Thom et al. (2017); Ponson (2016); Thompson et al. (1994)) and numerically (e.g. Gjerden et al. (2013); Zapperi et al. (2005); Barabási and Stanley (1995); Nakano et al. (1995)). Still, our system size is two orders of magnitude larger than the discrete size of the system.

The size of our system is limited by the computational power available today. An increase in the system size implies in fact much longer times to reach the steady-state; the time needed to reach such a state scales as L^z , with system size L and $z \geq 1.5$, the dynamic exponent of the process at hand (Barabási and Stanley (1995), cf. Supplementary Figure 2 of supplemental material). The scaling of the computing time due to system size in molecular dynamics simulations must also be considered, and the overall simulation time thus scales approximately as $L^{2.5}$, clearly beyond practical.

3.3 Critical length scale and further remarks

3) Role of the "critical length scale": The authors argue "that the inclusion of the critical length scale for the ductile-to-brittle transition is essential to capture the evolution to the self-affine morphology" but this length scale does not appear to affect the outcome of their simulation. I agree that a ductile-to-brittle transition is necessary to explain the authors results. However, a length-dependent ductile-to-brittle transition is also documented elsewhere, e.g. in the indentation literature; this is not a new concept.

As the reviewer agrees, such a transition is necessary to explain our results, and we thus cited and briefly discussed some previous works. The choice of the citations fell on those most appropriate for the adhesive wear process, which we are investigating. In addition, while the ductile to brittle transition is certainly not a new concept in materials science, it is a new concept in surface topography evolution (in tribology and geophysics).

4) I find the name "molecular dynamics simulations" unfortunate for the authors model. As argued in my earlier report, there is no molecular scale in this model. "Discrete element method" or "mesoparticle method" would capture the essence better.

The reviewer argued in the earlier report that our model is not coarse-grained, thus the recommendation to call it "discrete element method" or "mesoparticle method" appears contradictory to us. DEM in particular is quite different, as it involves both normal and tangential springs, friction, and particles of finite radius and volume. We stick to the expression "molecular dynamics", as it is commonly accepted to indicate the category of discrete *numerical* methods with mass points interacting via some potential energy function. The widely used software LAMMPS (which we use), for instance, is a molecular dynamics simulator, but clearly several other physical entities besides molecules are investigated with it. Indeed, numerous "molecular dynamics" simulations do not in fact simulate molecules at all, but solids.

5) Langevin thermostat: By point was not on how the force is computed by that the Langevin damping affects the simulated dissipation. However, the NVE test is suitable to show that the influence of thermostating is negligible.

6) The authors still do not explain how the surface is extracted in the updated methods section. What is the value of Δx ? Is this the maximum atom in bins of size Δx ? Can more than one surface atom be in one bin?

We thank the reviewer for catching the oversight. Δx is approximately $1 r_0$. We updated the Methods section accordingly.

References

- A-L Barabási and Harry Eugene Stanley. *Fractal concepts in surface growth*. Cambridge university press, 1995.
- Knut S Gjerden, Arne Stormo, and Alex Hansen. Universality classes in constrained crack growth. *Physical Review Letters*, 111(13):135502, 2013.
- Binqian Luan and Mark O Robbins. Hybrid atomistic/continuum study of contact and friction between rough solids. *Tribology letters*, 36(1):1–16, 2009.
- Aiichiro Nakano, Rajiv K Kalia, and Priya Vashishta. Dynamics and morphology of brittle cracks: A molecular-dynamics study of silicon nitride. *Physical review letters*, 75(17):3138, 1995.
- Laurent Ponson. Statistical aspects in crack growth phenomena: how the fluctuations reveal the failure mechanisms. *International Journal of Fracture*, 201(1):11–27, 2016.
- Jean Schmittbuhl, Jean-Pierre Vilotte, and Stéphane Roux. Reliability of self-affine measurements. *Physical Review E*, 51(1):131, 1995.
- CA Thom, EE Brodsky, RW Carpick, GM Pharr, WC Oliver, and DL Goldsby. Nanoscale roughness of natural fault surfaces controlled by scale-dependent yield strength. *Geophysical Research Letters*, 44(18):9299–9307, 2017.
- C Thompson, Georgios Palasantzas, YP Feng, SK Sinha, and J Krim. X-ray-reflectivity study of the growth kinetics of vapor-deposited silver films. *Physical Review B*, 49(7):4902, 1994.
- Stefano Zapperi, Phani Kumar VV Nukala, and Srđan Šimunović. Crack roughness and avalanche precursors in the random fuse model. *Physical Review E*, 71(2):026106, 2005.

REVIEWERS' COMMENTS:

Reviewer #3 (Remarks to the Author):

My main concern, namely the data does not support the authors' conclusions on self-affinity, remains unresolved.

My main concern is that the statistics of the authors' data is not sufficient to draw the conclusions in the manuscript. I raised the issue that the rms height of the interface is too small; there are only a few "steps" on the surface and this leads to the discrete height distribution shown in Fig. R.1 in the rebuttal.

As a response, the authors cite literature where similar analyses have been carried out on systems of similar size. Let us for example look at PRE 71, 026106 (Zapperi, Nukala and Simunovic). While it is correct that the lateral size is comparable, rms height of the interface is much larger. This leads to the continuous distributions shown in Fig. 5 of that paper that can be collapsed as a function of size accordingly. No such collapse would be possible for the distributions obtained by the authors.

I therefore maintain that this paper should not be published with the present conclusions.

Reviewer #4 (Remarks to the Author):

The manuscript analyzes the roughness of two dimensional surfaces subject to friction and wear by molecular dynamics simulations. The key issue is the development of a self-affine surface. The previous referees have commented at length on several aspects of the manuscript and the authors responded in details. At this stage of the referee process, I want to express my opinion on the outstanding issue of the reliability of the roughness exponent evaluation. I think that the analysis presented is up to the current standards in the field. The authors employ two different methods, in real and Fourier space, and the result they get is consistently an exponent larger than $H=0.5$ and around $H=0.7$. It is true that there are not many decades of scaling, as pointed out by one of the referees, but I found the result to be convincing.

The authors might want to improve the presentation by binning the curves, for instance using log-spaced bins (and reporting the current graphs in the supplement). Another additional test would be to compute the local log-slope of the curves to show that possible trends are not large.

REVIEWERS' COMMENTS

We are thankful to reviewers 3 and 4 for investing their time and expertise in the review process.

The only remaining point of disagreement with reviewer 3 is that the height distributions only consist of "a few steps" and are insufficient for the presented analysis. We have shown that increasing the simulation box size, the rms roughness increases as expected for self affine surfaces. Unfortunately it is not computationally affordable to reach simulations sizes that would fully convince reviewer 3. Nonetheless, we reiterate that our results soundly reveal that the Hurst exponent is above 0.5, i.e. the formation of roughness is not a random process but it is controlled by plastic and fracture events at the junction between debris and surfaces. We are sorry we could not convince reviewer 3.

We are grateful that reviewer 4 supports our work (as did reviewers 1 and 2 previously) and assesses that the analysis is up to the current standards in the field and is convincing. As for the binning of the curves, we prefer to keep the plots as they are and show the fluctuations in the data.

Reviewer #3 (Remarks to the Author):

My main concern, namely the data does not support the authors' conclusions on self-affinity, remains unresolved.

My main concern is that the statistics of the authors' data is not sufficient to draw the conclusions in the manuscript. I raised the issue that the rms height of the interface is too small; there are only a few "steps" on the surface and this leads to the discrete height distribution shown in Fig. R.1 in the rebuttal.

As a response, the authors cite literature where similar analyses have been carried out on systems of similar size. Let us for example look at PRE 71, 026106 (Zapperi, Nukala and Simunovic). While it is correct that the lateral size is comparable, rms height of the interface is much larger. This leads to the continuous distributions shown in Fig. 5 of that paper that can be collapsed as a function of size accordingly. No such collapse would be possible for the distributions obtained by the authors.

I therefore maintain that this paper should not be published with the present conclusions.

Reviewer #4 (Remarks to the Author):

The manuscript analyzes the roughness of two dimensional surfaces subject to friction and wear by molecular dynamics simulations. The key issue is the development of a self-affine surface. The previous referees have commented at length on several aspects of the manuscript and the authors responded in details. At this stage of the referee process, I want to express my opinion on the outstanding issue of the reliability of the roughness exponent evaluation. I think that the analysis presented is up to the current standards in the field. The authors employ two different methods, in real and Fourier space, and the result they get is consistently an exponent larger than $H=0.5$ and around $H=0.7$. It is true that there are not many decades of scaling, as pointed out by one of the referees, but I found the result to be convincing.

The authors might want to improve the presentation by binning the curves, for instance using log-spaced bins (and reporting the current graphs in the supplement). Another additional test would be to compute the local log-slope of the curves to show that possible trends are not large.

Stefano Zapperi